

# Hidden interactions in the intertidal rocky shore: variation in pedal mucus microbiota among marine grazers that feed on epilithic biofilm communities

Clara Arboleda-Baena[1,2], Claudia Belén Pareja[2], Isadora Pla[2], Ramiro Logares[3], Rodrigo De la Iglesia[2,4] and Sergio Andrés Navarrete[1,4,5]

[1] Estación Costera de Investigaciones Marinas and Center for Applied Ecology and Sustainability (CAPES), Pontificia Universidad Católica de Chile, El Tabo, Chile
[2] Departamento de Genética Molecular y Microbiología, Pontificia Universidad Católica de Chile, Santiago de Chile, Región Metropolitana, Chile
[3] Institut de Ciències del Mar (ICM), CSIC, Barcelona, Catalonia, Spain
[4] Marine Energy Research & Innovation Center (MERIC), Santiago de Chile, Chile
[5] Centro Basal COPAS-COASTAL, Universidad de Concepción

Corresponding author
Clara Arboleda-Baena,
claraarboledab@gmail.com

## ABSTRACT

In marine ecosystems, most invertebrates possess diverse microbiomes on their external surfaces, such as those found in the pedal mucus of grazing gastropods and chitons that aids displacement on different surfaces. The microbes are then transported around and placed in contact with free-living microbial communities of micro and other macro-organisms, potentially exchanging species and homogenizing microbial composition and structure among grazer hosts. Here, we characterize the microbiota of the pedal mucus of five distantly related mollusk grazers, quantify differences in microbial community structure, mucus protein and carbohydrate content, and, through a simple laboratory experiment, assess their effects on integrated measures of biofilm abundance. Over 665 Amplicon Sequence Variants (ASVs) were found across grazers, with significant differences in abundance and composition among grazer species and epilithic biofilms. The pulmonate limpet *Siphonaria lessonii* and the periwinkle *Echinolittorina peruviana* shared similar microbiota. The microbiota of the chiton *Chiton granosus,* keyhole limpet *Fissurella crassa,* and scurrinid limpet *Scurria araucana* differed markedly from one another, and form those of the pulmonate limpet and periwinkle. Flavobacteriaceae (Bacteroidia) and Colwelliaceae (Gammaproteobacteria) were the most common among microbial taxa. Microbial strict specialists were found in only one grazer species. The pedal mucus pH was similar among grazers, but carbohydrate and protein concentrations differed significantly. Yet, differences in mucus composition were not reflected in microbial community structure. Only the pedal mucus of *F. crassa* and *S. lessonii* negatively affected the abundance of photosynthetic microorganisms in the biofilm, demonstrating the specificity of the pedal mucus effects on biofilm communities. Thus, the pedal mucus microbiota are distinct among grazer hosts and can affect and interact non-trophically with the epilithic biofilms on which grazers feed, potentially leading to microbial community coalescence mediated by grazer movement. Further studies are needed to unravel the myriad of non-trophic interactions and their reciprocal impacts between macro- and microbial communities.

# INTRODUCTION

'Out of sight, out of mind' has been the approach most ecologists, terrestrial and marine, have followed when it comes to understanding the role of species interactions on the functioning of ecosystems and communities, disregarding microscopic organisms as sufficiently 'isolated' from their coexisting macroscopic components. Several exceptions to this oversight abound in the literature (*Wahl et al., 2012*; *Rosenberg & Zilber-Rosenberg, 2016*), indeed. The discovery that macroscopic organisms host unique and diverse assemblages of microorganisms, the microbiomes, transformed our understanding of animal biology, ecology, and evolution (*Lafferty et al., 2008*; *McFall-Ngai et al., 2013*) and it certainly accelerated research on the connections and dependencies between macroscopic and microscopic worlds. But these advances have been largely biased towards internal tissue microbiomes, mostly in mammals, and especially in humans (*Costello et al., 2012*; *Tremaroli & Bäckhed, 2012*). In marine ecosystems, the study of susceptibility to macroscopic biofouling on artificial surfaces covered by microbial biofilms at sea has been an active focus of research from an applied material science perspective (*Salta et al., 2010*; *Navarrete et al., 2019*; *Navarrete et al., 2020*; *Daille et al., 2020*; *Antunes et al., 2020*). Still, in most natural systems, our understanding of the macroscopic and microscopic interactions remains rudimentary, at best.

The intensively studied rocky shore communities represent a model system that can help us disentangle the complex networks of interactions between microbes and the co-occurring macroscopic invertebrates and macroalgae (*Hawkins et al., 2008*; *Kéfi et al., 2015*). Pioneering studies on many rocky shores have documented trophic interactions between invertebrates and microbial biofilms (*Nicotri, 1977*; *Underwood, Denley & Moran, 1983*; *Hill & Hawkins, 1991*; *Thompson et al., 2000*). However, consumption is but one of the diverse types of interactions that can occur between these worlds. Non-trophic interactions are probably as important and diverse, and some of those interactions have no similar counterpart in the macroscopic world (*Kéfi et al., 2012*). This is the case of microbiomes of macroscopic organisms, which interact with microorganisms found, for instance, in the epilithic biofilms. Interactions between these microscopic communities, mediated by grazers, could produce wholesale exchanges of species at all trophic levels, also known as community coalescence (*Rillig et al., 2015*), but in this case, occurring frequently and extensively as the grazers move about the shore.

Recent studies have described the microbial species composition and some of the functional roles of microbiomes of marine gastropods and polyplacophorans, including gut ducts (*Dudek et al., 2014*; *Aronson, Zellmer & Goffredi, 2016*; *Cicala et al., 2018*), gill cells (*Zbinden et al., 2015*), the outer body surface (*Fukunaga et al., 2008*; *Davis et al.,*

*2013*), and microbiome changes across marine biogeographic boundaries (*Neu, Allen & Roy, 2019*).

Pedal mucus, which is essential for animal motility in all mobile gastropods and chitons (*Denny, 1980*) and is secreted by the pedal gland located inside the front end of the foot (*Davies & Hawkins, 1998*), is in direct contact with the rock surface and biofilm microbial communities, yet its microbiome has not been studied in detail. As the organism moves about grazing on the shore, some of the onboard microbial communities of the foot are placed in direct contact with the different components (*e.g.*, species) of biofilm communities found on the rock surface, potentially exchanging species and interacting in ways that could modulate the grazer pedal mucus microbiomes as well as biofilm diversity, composition, and functional attributes. Mollusk pedal mucus is composed of about 96% of water and the ∼4% has different proportion of proteins, carbohydrates, lipids glycoproteins (*Denny, 1980*; *Davies, Hawkins & Jones, 1990*), and mineral salts (*Shashoua & Kwart, 1959*).

Pedal mucus can have a negative effect on biofilms, like antibacterial activity, described in the mucus trail of the predatory whelk *Achatina fulica* over both Gram-negative and Gram-positive cultures (*Iguchi, Aikawa & Matsumoto, 1982*). Similarly, the pedal mucus of the grazing pulmonate limpet, *Siphonaria pectinata,* contains the pectinatone antibiotic, which acts against Gram-positive bacteria like *Staphylococcus aureus, Bacillus subtilus*, and fungi like *Candida albicans,* and *Saccharomyces cerevisiae* on cultured experiments (*Biskupiak & Ireland, 1983*). It has been suggested that antibacterial activity in the mucus is probably associated with the protein and polypeptide concentration, not the carbohydrates, due to higher presences of enzymes and/or pyrone derivative in the pedal mucus (*Iguchi, Aikawa & Matsumoto, 1982*; *Biskupiak & Ireland, 1983*).

In contrast, it has been shown that microorganisms and early stages of macroalgae can settle and grow faster on the mucus trail of intertidal grazers (*Littorina peruviana, Tegula atra, Siphonaria lessonii,* and *Collisella* sp.) than on clean rock surfaces along the central coast of Chile (*Santelices & Bobadilla, 1996*). Similar positive effects on microalgal growth were described for the pedal mucus of the limpets *Lottia gigantea, Collisela digitalis,* and *Collisela scabra* on the Pacific shore of North America (*Connor, 1986*). Increased colonization of heterotrophic microorganisms was stimulated by the pedal mucus of the turbinid snail *Monodonta turbinata*, suggesting the mucus provides organic enrichment for microbial growth (*Herndl & Peduzzi, 1989*). That is also the case for the pedal mucus of the small abalone, *Haliotis diversicolor*, which stimulates the growth of bacteria *Escherichia coli* and *Staphylococcus epidermidis* (*Guo et al., 2009*). Biochemical analyses suggest that the ability of mucous trails to trap microalgae adhesively and to stimulate microalgae growth is correlated with carbohydrate content (*Connor, 1986*).

Thus, the pedal mucus of mobile mollusks can affect epilithic microbial communities. The effects can be positive or negative on some microorganisms, which may be related to the mucus chemical (protein/carbohydrate concentration) or microbiome composition. Identifying a core microbiota, common members to two or more microbial assemblages associated with a habitat (*Turnbaugh et al., 2007*; *Hamady & Knight, 2009*; *Shade & Handelsman, 2012*) is the first step in defining pedal mucus communities, understanding

responses to perturbation, and the components that are resilient and persistent across microbial assemblages (*Shade & Handelsman, 2012*).

The wave-exposed rocky shores of central Chile are characterized by a diverse assemblage of molluskan grazers that belong to different orders and families and co-occur closely (*Santelices, Vásquez & Meneses, 1986*; *Rivadeneira, Fernández & Navarrete, 2002*; *Aguilera & Navarrete, 2011*; *Aguilera & Navarrete, 2012*). Although their impacts on macroalgal communities and ecological succession can be quite different (*Aguilera & Navarrete, 2012*; *Aguilera, Navarrete & Broitman, 2013*; *Aguilera et al., 2020*), they overlap in their diets (*Santelices, Vásquez & Meneses, 1986*; *Camus, 2008*; *Camus, Arancibia & Ávila Thieme, 2013*) and all feed regularly on microbial biofilms (*Camus et al., 2009*; *Aguilera, Navarrete & Broitman, 2013*; *Kéfi et al., 2015*). In this model ecosystem, we characterized the pedal mucus microbiota of five common, but distantly related, intertidal mollusks of central Chile as a first step in understanding potential non-trophic interactions between grazers and epilithic biofilms. Using an array of molecular techniques and replicated experiments, we evaluated two general hypotheses: (1) that because all these grazers co-occur on the same wave-exposed rocky shore habitat and consume microbial biofilm, they all will share similar microbial communities in the microbiota of the pedal mucus, with same core microbial groups dominating both epilithic biofilms and grazer microbiota, (2) since the intertidal grazers species may have differences in the pedal mucus content of protein/carbohydrates, these will correlate with differences in their pedal mucus microbiota composition, and in the effects of pedal mucus on the abundance of photosynthetic biofilm.

## MATERIALS & METHODS

### Grazer assemblage

We characterized the intertidal mollusk grazer assemblage of the wave exposed shores of Las Cruces, central Chile (33°30′S, 71°38′W) during December 2017 and January 2018. Gently inclined (∼40°) rocky platforms were chosen inside the marine reserve of the Estación Costera de Investigaciones Marinas, Pontificia Universidad Católica de Chile, from which fishermen have been excluded since 1982 (*Castilla & Durán, 1985*; *Castilla, 1999*; *Navarrete, Gelcich & Castilla, 2010*) and similarly exposed platforms outside the reserve, roughly 145.9 m to the south, where fishers collect some of the mollusk species. At the high, mid, and low intertidal zones (following *Castilla, 1981*; *Fernández et al., 2000*), we haphazardly laid down ten $50 \times 50$ cm quadrats ($0.25$ m$^2$) along a 15 m long transect parallel to the shoreline. All grazers inside quadrats were measured (maximum length). We estimated the biomass (g m-2) of each species through the wet-weight average per m-2 (see data in Figshare: https://doi.org/10.6084/m9.figshare.15113490.v2). We chose five of the most abundant species in terms of total biomass (Fig. 1): one Polyplacophoran, the chiton *Chiton* granosus Frembly, 1828 (Family Chitonidae), and four Gastropods, the Littorinid *Echinolittorina peruviana* Lamarck, 1822 (Family Littorinidae), the keyhole limpet *Fissurella crassa* Lamarck, 1822 (Family Fissurellidae), the scurrinid limpet *Scurria araucana* d'Orbigny, 1839 (Family Lottiidae), and the pulmonate limpet *Siphonaria lessonii* Blainville, 1827 (Family Siphonariidae) (*Espoz et al., 2004*; *Aguilera,*

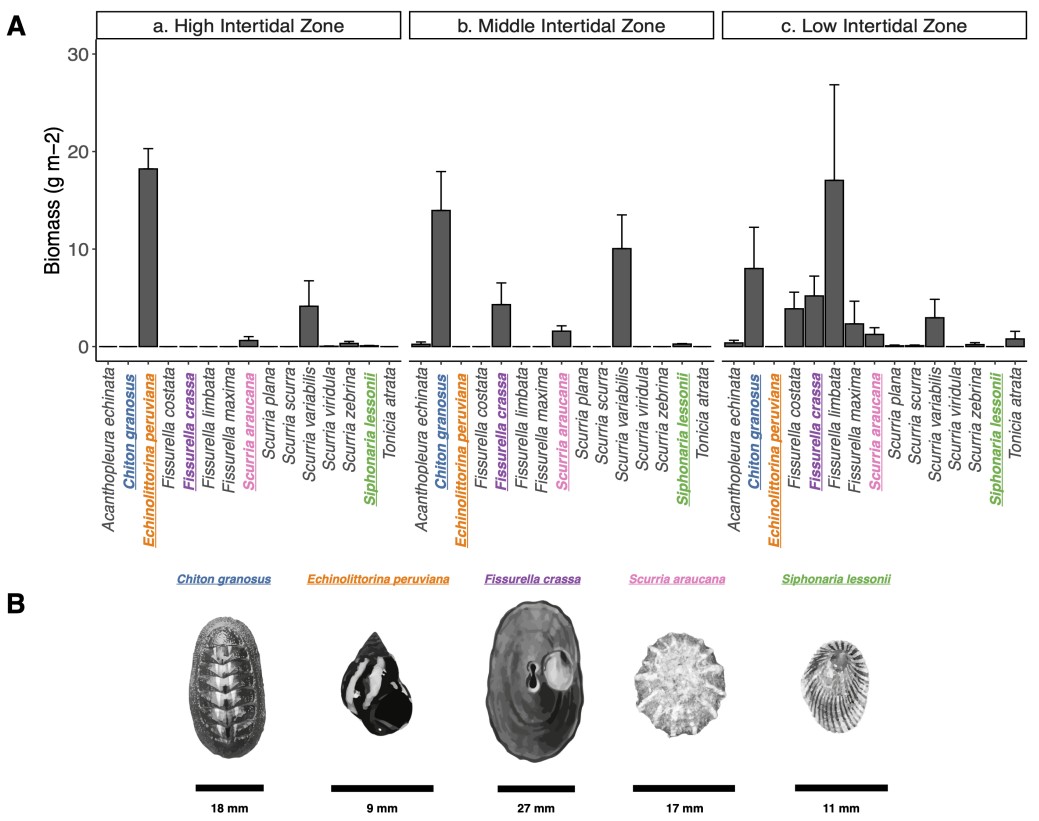

**Figure 1** **Grazer assemblage biomass at the intertidal rocky shore of central Chile.** (A) Grazer assemblage biomass (g m$^{-2}$) at the intertidal rocky shore of central Chile. (a) High, b) Middle, and c) Low intertidal zones (mean + SE). (B) Species selected for analysis are highlighted in colors: (blue) Chiton granosus, (orange) Echinolittorina peruviana, (purple) Fissurella crassa, (pink) Scurria araucana, (green) Siphonaria lessonii. Scale bar is shown for each species.

*Navarrete & Broitman, 2013*; *Camus, Arancibia & Ávila Thieme, 2013*). These five species are evolutionarily distantly related and can co-occur closely in wave exposed platforms and other intertidal microhabitats. They are classified as ecological trophic omnivores, scraping the rock surfaces, removing spores, macroalgae seedlings, epiphytes, microorganisms (periphyton and epilithic biofilm) and also newly established invertebrates (*Santelices, Vásquez & Meneses, 1986*; *Aguilera & Navarrete, 2007*; *Aguilera & Navarrete, 2011*; *Aguilera & Navarrete, 2012*; *Camus, 2008*).

## Pedal mucus microbiota analyses

The study was conducted at the Estación Costera de Investigaciones Marinas (ECIM) of Pontificia Universidad Católica de Chile (PUC), at Las Cruces, Valparaíso, Chile: 33o 30ʹS, 71o 38ʹ. Field study approval number ID Protocol: 170829006, by the Comité Institucional de Seguridad en Investigación of the Pontificia Universidad Católica de Chile. We collected ten animals of each species from gently sloping wave-exposed platforms nearby the marine reserve of Las Cruces, focusing on the intertidal zone they were more abundant (Fig. 1). All animals were collected during nocturnal low tides to prevent foot damage (*Aguilera &*

*Navarrete, 2011*; *Aguilera & Navarrete, 2012*), and the shell and foot area were measured (Supplement 1). Collected individuals were brought to the laboratory in coolers and then placed to acclimatize for a week in separate aquaria to reduce animal stress, with the same source of circulating seawater and constant aeration. There, grazers could only feed on epilithic biofilm that was provided in the same manner to all individuals. To minimize contamination from grazers' feces, two days before the experiment, individuals were "cleaned" by placing them in an aquarium with constant aeration and 0.2 μm filtered seawater (taken from the same location). The two day period was long enough to reduce feces during the experiment and short enough to avoid locomotory and metabolic adverse effects (*Calow, 1974*), because on the third day animals reduced their motility (C Arboleda-Baena, 2019, per. obs.). The sterile water was replaced every 2–6 h to minimize ammonium concentration and prevent biofilms formation associated with animals' feces. We expect that this cleaning period also helped remove microorganisms that are incidental on the animal foot and do not maintain populations in the pedal mucus. The motility and behavior of the animals were monitored throughout the acclimation period. Seawater temperature was maintained at 13 °C $\pm$ 2 °C, which was the average SST during the time of experiments. To obtain samples of the pedal mucus microbiota, five grazers of each species were chosen randomly and placed in individual clean aquaria (14.3 ×14.3 × 12.5 cm) with 400 ml filtered seawater (0.22 μm filtered and taken from the same location). Five treatments (a-e) with 5 replicates each were applied for mollusk pedal mucus collection. The experimental unit had cover glass slides of 75 × 25 × 1.5 mm previously autoclaved and placed in the bottom of the aquaria. In these individual arenas we included one individual of (a) *Chiton granosus,* (b) *Echinolittorina peruviana*, (c) *Fissurella crassa*, (d) *Scurria araucana*, or (e) *Siphonaria lessonii*. We used as control five aquaria with natural epilithic biofilms grown on granitic rock subjected to the same source of circulating seawater where the grazers were acclimatized. Rocks were collected from the field, cut into coupons of 3 × 8 × 2 cm with a COCH Bridge saw machine that prevented overheating and potential mineral modification, and cleaned by deionized water, dried, maintained at room temperature before introducing them in experimental aquaria. The biofilm growing on the rock surface without grazers was used as a reference for free living epilithic biofilms against which we compare grazer microbiota. Naturally growing biofilms vary greatly in composition and structure depending on successional stage and a suit of microclimatic and environmental conditions (*Dang & Lovell, 2016*). We therefore opted to use a single common reference for epilithic biofilm from the same source of circulating seawater where the grazers were acclimatized, *i.e.*, for all grazers microbiota.

Treatments were randomly assigned to the 30 experimental units (five replicates per treatment). Every three hours, temperature was checked, and feces were removed. The experiment lasted 24 h, and then the animals were carefully removed. Within 24 h, rocks and cover glass slides of all treatments were sonicated separately (*Morris, Monier & Jacques, 1998*; *Bjerkan, Witsø& Bergh, 2009*), filtered through 0.22 μm pore filters of hydrophilic polyether sulfone (Merck), and preserved in liquid nitrogen at −196 °C for later molecular analyses. During the microbiota analyses, we lost one sample of each of the following treatments due to a poor-quality Illumina sequencing run: *C. granosus, E.*

*peruviana, S. lessonii,* and epilithic biofilm control (Supplement 1). The largest mean foot area of individuals used in mucus collection was 23.05 ± 3.46 cm2 for *F. crassa* and the lowest value was for *S. lessonii* with 0.15 ± 0.02 cm2. The mean rock area analyzed was 24.84 ± 1.89 cm2 (Supplement 1).

## DNA extraction and 16S rRNA-gene sequencing

DNA extraction from filters was conducted with the Phenol-Chloroform method (*Fuhrman et al., 1988*). DNA concentration was measured with the Qubit HS dsDNA Assay kit in a Qubit 2.0 Fluorometer (Life Technologies, Carlsbad, CA, USA) according to manufacture protocols. The V4-V5 region of the 16S rRNA gene was amplified with the primers 515FB: GTGYCAGCMGCCGCGGTAA and 926R: CCGYCAATTYMTTTRAGTTT (*Quince et al., 2011*; *Parada, Needham & Fuhrman, 2016*). Amplicons were sequenced in a MiSeq Illumina platform (2 ×300 bp). Both PCR and sequencing were done at the Dalhousie University CGEB-IMR (https://imr.bio/). Sequence data were deposited at the European Nucleotide Archive (ENA) database under accession number PRJEB41739.

## Community analyses

Amplicon reads were analyzed using the DADA2 pipeline (*Callahan et al., 2016*; *Lee, 2019*) to characterize Amplicon Sequence Variants (ASVs) (*Callahan, McMurdie & Holmes, 2017*) that were used as a proxy of microbial species or Operational Taxonomic Units (OTUs). All graphics and statistical analyses were carried out in R with RStudio interface (*Racine, 2012*; *R Core Team, 2013*). Most community ecology analyses were carried out using the packages vegan v2.5-6 (*Oksanen et al. , 2015*) and phyloseq v1.30.0 (*McMurdie & Holmes, 2013*). Two rarefaction curves were generated, the first one with a fixed sampling effort of 1,869 reads per sample due to the size of the smallest dataset from one replicate of the epilithic biofilm control (Supplement 1Fig. S1). This dataset was used to compare between grazer microbiota and microbial communities of epilithic biofilms. The second rarefaction curves, with a fixed sampling effort of 10,599 reads per sample due to the size of the smallest dataset from one replicate of *F. crassa* microbiota, was used to compare between grazer microbiotas (Supplement 1, Fig. S2). To examine the microbial beta diversity of the grazer species (*i.e.*, pedal mucus) and epilithic biofilm, we used non-metric multidimensional scaling (NMDS) ordination, based on Bray–Curtis and Jaccard dissimilarities. To test for statistically significant differences in composition among the microbiota, we conducted a permutational analysis of variance (PERMANOVA) (*Anderson & Walsh, 2013*). To determine which treatment differed from others, we conducted pairwise post-hoc tests with False Discovery Rate (FDR) correction (*Benjamini & Hochberg (1995)*. We performed PERMDISP to test for differences in dispersions between groups (*Anderson, 2006*).

To compare richness and diversity among treatments we conducted separate one-way ANOVAs on richness and Shannon index, after inspection for normality and homoscedasticity, considering treatment (grazer species and control) as a fixed factor and used Tukey's post hoc test to establish the pattern of differences.

We defined microbe (ASV) habitat specialists and generalists according to the number of grazer species' microbiota they inhabit. To reduce the importance of rare ASVs, we
omitted ASVs with abundances below 100 reads (*Logares et al., 2013*) in the dataset with a fixed sampling effort of 10,599 reads. To quantify specificity to different pedal mucus microbiota, we calculated the Indicator Value (IndVal) (*Dufrene & Legendre, 1997*) with the labdsv package (*Roberts & Roberts, 2016*) as $IndVal_{ij} = Specificity_{ij} * Fidelity_{ij} * 100$. Where the $Specificity_{ij}$ is the proportion of samples of type "j" that contain an ASV "i" and the $Fidelity_{ij}$ is the proportion of the number of reads (abundance) of an ASV "i" that are in a "j" type of samples (*Dufrene & Legendre, 1997*).

## Pedal mucus protein and carbohydrate concentration

We conducted protein and carbohydrates analyses by collecting, during nocturnal low tides, seven animals of each species from the same platforms as those collected for experiments. Individuals were brought to the laboratory in coolers and then placed in separate aquaria with circulating seawater and constant aeration for one week. This time period reduces animal stress and allowed animals to acclimate to lab conditions. Then, collection of pedal mucus was done under a laminar flow cabinet (*Connor, 1986*). Animals were carefully removed from their containers, washed in filtered seawater (0.2 µm pore-size filters), and then placed individually on one inclined sterile glass slides (21 × 7 cm). Seven individuals of each species were placed in equal number of glass slides. Filtered seawater was added to stimulate movement and mucus production as the animal moved across de glass surface. They were removed from de glass after a maximum of 5 min of movement. To confirm the mucus pedal of the grazers was successfully collected, we used two of the glass slides ($n = 2$) and one of them was stained with Gram method (*Beveridge, Lawrence & Murray, 2007*) to detect gram-positive or gram-negative bacteria, and the other slide was stained with 0.01% acridine orange to stain DNA and RNA of the mucus microbial community (Supplement 2).

The mucus on the other five replicate glass slides was removed with a sterile scalpel and put it in individual cryovials with filtered seawater. Five cryovials were used to determine the mucus carbohydrate concentration ($n = 5$) with the phenol-sulfuric acid method (*Masuko et al., 2005*) and the protein concentration ($n = 5$) with the Bradford method (*Bradford, 1976*). Data were log-transformed and, since carbohydrates and protein are correlated, we tested for significant differences among carbohydrate and protein concentration among species using ANOVAs separately (Tukey post hoc test for heterogeneous variances was performed) and MANOVA, considering grazer species as a fixed factor. When a significant overall effect was detected, a Linear Discriminant Analysis (LDA) was used to cluster the different pedal mucus of the grazers by the carbohydrate and protein content.

Finally, to measure pedal mucus pH, we used five additional animals of each species, collected from the same platforms. Animals were brought to the laboratory in coolers and the foot pH was measured with a MQuant® pH test strips (resolution: 1.0 pH unit). These pH data were log-transformed ($n = 5$) to improve normality and then compared among grazer species using a Welch's ANOVA because slight heteroscedasticity remained after transformation. We considered grazers species as a fixed factor. Then, a Games-Howell post hoc test for heterogeneous variances was performed.

## Pedal mucus effects on integrated measures of epilithic biofilms abundance

To get a preliminary assessment of the effect of pedal mucus on integrated measures of the free-living biofilm community, we conducted a replicated laboratory experiment at ECIM to quantify effects on the abundance of photosynthetic biofilm components. To this end, we collected 4 animals of each species from wave-exposed platforms and brought them to the laboratory as described above. Collection of pedal mucus was conducted as described above.

We cultured epilithic biofilms of the intertidal rocky shore and then we took and placed biofilm inoculum on 24 cover glass slides inside a Polycarbonate cell culture plate of 6-Wells, for one week in K medium (*Keller et al., 1987*). The experiment consisted of placing the pedal mucus collected from the four individuals of each of the five species, in separate replicated wells with cover glass slides that had been cultured with biofilm. Four control wells received no mucus. Treatments were randomly assigned to the cell culture plates. After one week, we stained the cover glass slides with 200 µL 0.01% acridine orange (*Rigler, 1966*) for 5 min, and then we removed the biofilm. After 3 min incubation in the dark, the staining solution was removed, and the plate was washed twice with 500 µL of PBS solution. We took five photographs of each cover glass slide under a Fluorescent Carl filter 525/50 nm. Then, from the photographs, we measured the cover of the Zeiss AXIO Scope A1 Microscope using excitation filter (FS38) 470/40 nm and emission photosynthetic epilithic biofilm using the software Image J (*Arboleda-Baena et al., 2022*).

Covers under different treatments ($n = 20$) were analyzed with one-way ANOVA with grazer species as a fixed factor. A Tukey post-hoc test was performed to determine the pattern of differences.

## RESULTS

### Microbiota and epilithic biofilm microbial communities

We obtained 683,494 good-quality sequences from 26 samples. After rarefying to 1,869 reads per sample, due to the size of the smallest dataset from one epilithic biofilm sample, we had a total of 958 ASVs from all treatments (Supplement 1), of which 666 ASVs were found in the pedal mucus microbiota of the five grazers and the others in the epilithic biofilm community. Sequences from more than 17 Phyla were found in all mucus microbiota. Proteobacteria (Alphaproteobacteria and Gammaproteobacteria), Bacteroidetes, and Verrucomicrobia sequences were the most abundant and accounted for more than 90% of the reads (Fig. 2). The pedal mucus microbiota of the grazers *C. granosus, F. crassa,* and *S. araucana* were dominated by Bacteroidetes sequences (66.5%, 67.3%, and 49.9% relative abundance, respectively); while the pedal mucus microbiota of *E. peruviana* and *S. lessonii* were dominated by Gammaproteobacteria (66.2% and 77.5% of their total reads, respectively), and Epsilonbacteraeota (14.7% and 6.3%, respectively) sequences that were absent in the other three species (Table S1 in Supplement 3). In contrast, in the epilithic biofilm, Alphaproteobacteria, Gammaproteobacteria, Bacteroidetes, Planctomycetes and Verrucomicrobia sequences were dominant (Fig. 2), but the most abundant were

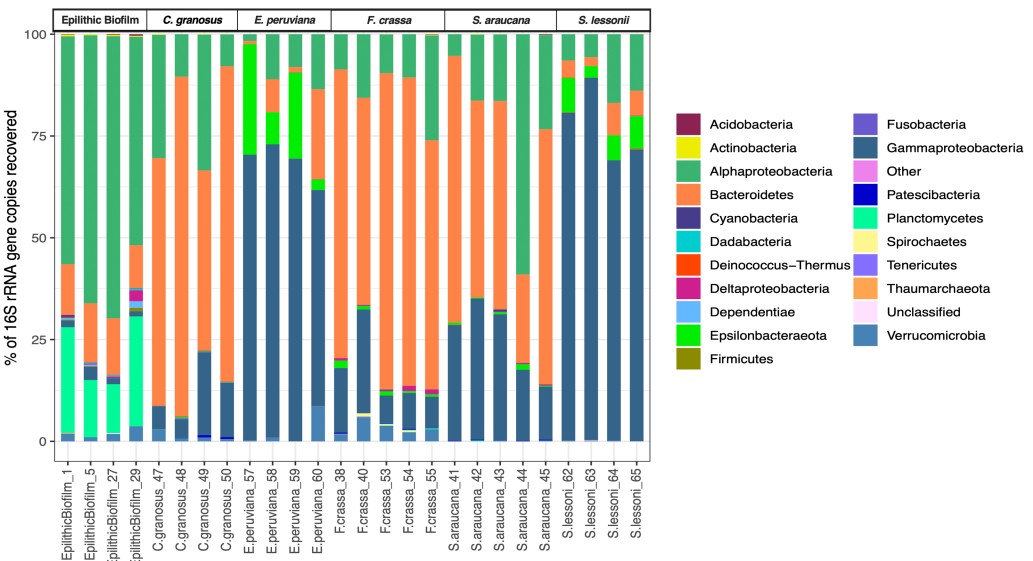

**Figure 2** **Microbial community relative abundance composition (Bacteria and Archaea) of pedal mucus microbiota.** Relative abundance of microbial community composition (Bacteria and Archaea) of *Chiton granosus, Echinolittorina peruviana, Fissurella crassa, Scurria araucana, Siphonaria lessonii* pedal mucus microbiota, and epilithic biofilms.

Alphaproteobacteria (60.6% of total reads) and Planctomycetes (19.7% of total reads). The last was absent in the pedal mucus microbiota (Fig. 2; Table S1 in Supplement 3).

All mollusk species had, as the most abundant microorganisms sequences, the Family Flavobacteriaceae followed by the Colwelliaceae (Bacteroidetes and Gammaproteobacteria respectively) (Table S2 in Supplement 3). The microbial communities of all species were different, with low dispersion of composition within *F. crassa* and *C. granosus*, and a large dispersion in *S. araucana* (Fig. 3). There was only high community overlap between the microbiota of *E. peruviana* and *S. lessonii* (Fig. 3). Analyses of the epilithic biofilm showed large and significant differences with the pedal mucus microbiota of all species (Supplement 4). Differences in microbial communities among grazer species and epilithic biofilms were statistically significant with Bray-Curtis and Jaccard distances (PERMANOVA, $df = 5$, $p = 0.0009$), and FDR-adjusted post-hoc tests showed that all species differed from each other, except *E. peruviana* and *S. lessonii* ($p$ adjusted = 0.9, for both distances. Tables S1 and S2 in Supplement 4). We performed the same analyses with the rarefaction level dataset of 10,599 reads and compared the similarity between grazer microbiotas. We obtained the same results with Bray–Curtis and Jaccard distances, differences that were statistically significant between grazers microbiota (PERMANOVA, $df = 5$, $p = 0.0009$), and the FDR-adjusted post-hoc test showed no differences between the microbial communities of *E. peruviana* and *S. lessonii* ($p$ adjusted = 0.9, for both distances. Tables S3 and S4 in Supplement 4). After PERMDISP analysis, we proved, with an ANOVA test, the groups' dispersions were not different for both rarefaction datasets (Figs. S3 and S4 in Supplement 4).

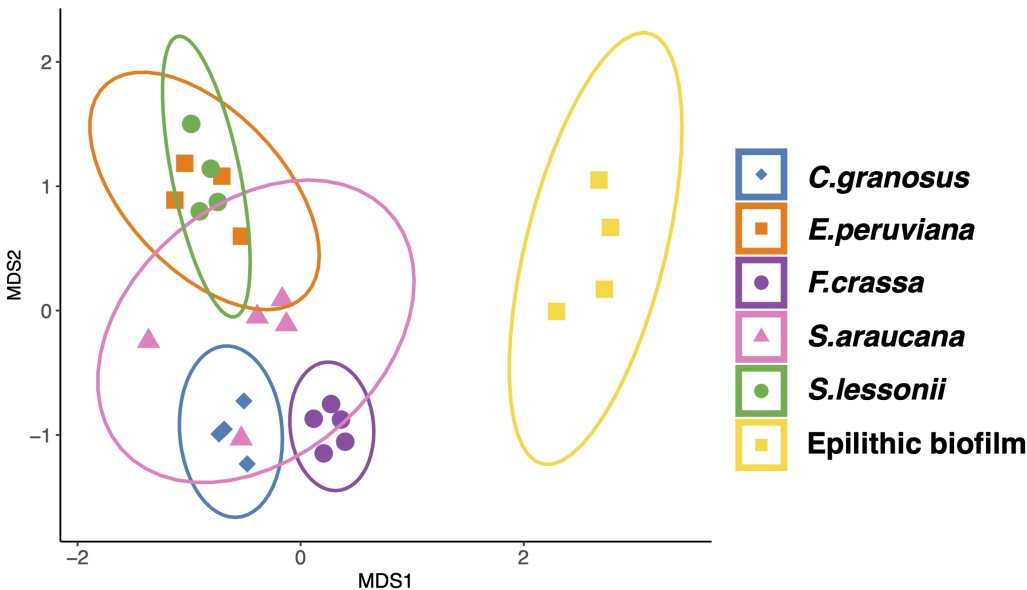

**Figure 3** **Microbiota compositional similarity of the most abundant grazers of the intertidal rocky shore and the epilithic biofilm.** Non-metric multidimensional scaling (NMDS) ordination plots based on Bray–Curtis distances. The shapes denote the microbiota grazer species surrounded by an ellipse showing the 95% confidence interval: (♦) *Chiton granosus*, (■) *Echinolittorina peruviana*, (●) *Fissurella crassa*, (▲) *Scurria araucana*, (●) *Siphonaria lessonii*, and (■) Epilithic biofilm. Stress = 0.101.

We found significant differences in microbiota richness (mean + SE) among treatments (Fig. S1A in Supplement 5, ANOVA, $df = 5$, $p = 0.0001$), with similar values among *F. crassa* (138.6 ± 8.39), *C. granosus* (165.25 ± 14.97), *S. araucana* (130.6 ± 21.09) and epilithic biofilms (137.75 ± 29.37), and significantly lower richness in *E. peruviana* (67.25 ± 6.42) and *S. lessonii* (66.25 ± 6.54) (Table S1 in Supplement 6; Tukey post hoc tests). The Shannon diversity index (mean + SE) also showed significant differences among treatments (Supplement 5 Fig. S1B , ANOVA, $df = 5$, $p = 0.0002$), epilithic biofilm (4.22 ± 0.14), *C. granosus* (3.63 ±0.28), *E. peruviana* (2.77 ± 0.12), *F. crassa* (3.83 ± 0.06), *S. araucana* (3.20 ± 0.27), and *S. lessonii* (2.71 ± 0.17). However, it was not possible to resolve a clear pattern of difference between species pairs using the Tukey post-hoc test due to a lack of power (Table S2 in Supplement 6). Richness changed with a rarefaction level of 10,599 reads (Supplement 1 and Fig. S2 in Supplement 5). Nevertheless, differences between richness and diversity treatments maintain the same pattern observed for rarefaction level <2000 reads (See Fig. S2 in Supplement 5; Tables S1–S4 in Supplement 6).

After rarefying to 10,599 reads per sample, due to the size of the smallest dataset from one *F. crassa* sample, we had a total of 1,205 ASVs from all pedal mucus microbiota treatments (Supplement 1). The pedal mucus microbiome of *C. granosus* displayed an average of 236 ASVs, 107 ASVs for *E. peruviana*, 189 ASVs for *F. crassa*, 237 ASVs for *S. araucana,* and 112 ASVs for *S. lessonii* (Supplement 1). Our analyses of specificity

and fidelity at the microbial family level, which reduced the number of ASVs to 123 taxa, showed that the pedal mucus microbiota of *C. granosus, E. peruviana, F. crassa,* and *S. araucana* were composed mostly by Flavobacteriaceae sequences. *C. granosus* also has habitat specialists from Bacteroidetes (Crocinitomicaceae, Cryomorphaceae, Cyclobacteriaceae, and Saprospiraceae), Alphaproteobacteria (Rhodobacteraceae and Phyllobacteriaceae), Gammaproteobacteria (Alteromonadaceae) and Verrucomicrobia (Rubritaleaceae) sequences. In turn, *E. peruviana* has habitat specialist ASVs from Epsilonbacteraeota (Arcobacteraceae), Gammaproteobacteria (Colwelliaceae and Nitrincolaceae) and Verrucomicrobia (Rubritaleaceae) sequences. *F. crassa* has habitat specialists from Bacteroidetes (Crocinitomicaceae and Cyclobacteriaceae), Alphaproteobacteria (Rhodobacteraceae), Gammaproteobacteria (Nitrincolaceae, Cardiobacteriaceae, Pseudoalteromonadaceae, Vibrionaceae, and Shewanellaceae), Verrucomicrobia (Rubritaleaceae), Deltaproteobacteria (Bacteriovoracaceae) and Spirochaetes (Spirochaetaceae) sequences. *S. araucana* has habitat specialists from Alphaproteobacteria (Rhodobacteraceae) and Gammaproteobacteria (Nitrincolaceae) sequences. In the case of *S. lessonii*, the most abundant habitat specialist ASVs were Alphaproteobacteria (Rhodobacteraceae), but it also has ASVs from Gammaproteobacteria (Alteromonadaceae, Cellvibrionaceae and Marinomonadacea) sequences (Fig. 4; Supplement 7).

## Pedal mucus protein and carbohydrate concentration

The method developed to obtain pedal mucus from live individuals was successful in all cases (Supplement 7). We found statistical differences in carbohydrate content among mollusk pedal mucus (Fig. 5A, and Table S1 in Supplement 8, ANOVA, $df = 4$, $p = 0.0004$). The carbohydrate content values µg/ml (mean ± SE) in descending order were *F. crassa* (41.96 ± 8.59), *C. granosus* (11.49 ± 4.50), *E. peruviana* (6.28 ± 12.21), *S. lessonii* (2.69 ± 11.29) and *S. araucana* (2.65 ± 2.04). The protein content also varied among pedal mucus grazers (Fig. 5B, and Table S2 Supplement 8, ANOVA, $df = 4$, $p = 0.006$). The protein content values µg/ml (log mean ± SE,) were *F. crassa* (1.61 ± 0.4), *S. lessonii* (0.94 ± 0.35) ,), *E. peruviana* (0.51 ± 0.21), *C. granosus* (0.27 ± 0.27), and *S. araucana* (0 ± 0). Both mean carbohydrate and protein concentration varied significantly among grazer species (Fig. 5A and Fig. 5B, MANOVA, $df = 4$, $p = 0.018$). The bivariate plot of carbohydrate and protein content showed the pedal mucus of *F. crassa* had a much higher concentration of both components than that of all other species. In contrast, *S. araucana* had the lowest, with nearly nil protein content (Fig. 5D). To cluster the different pedal mucus of the grazers by the carbohydrate and protein content, we performed an LDA. This showed slightly different patterns with the limpets *F. crassa* and *S. lessonii* clustering together and on opposite bivariate ends than *S. araucana* (Supplement 9).

The mean pedal mucus pH ranged from 8.2 to 9.0 among species, with little variability among individuals within species or among species (Fig. 5C). In some cases (*C. granosus, E. peruviana*) no variation in pH was detected among individuals.

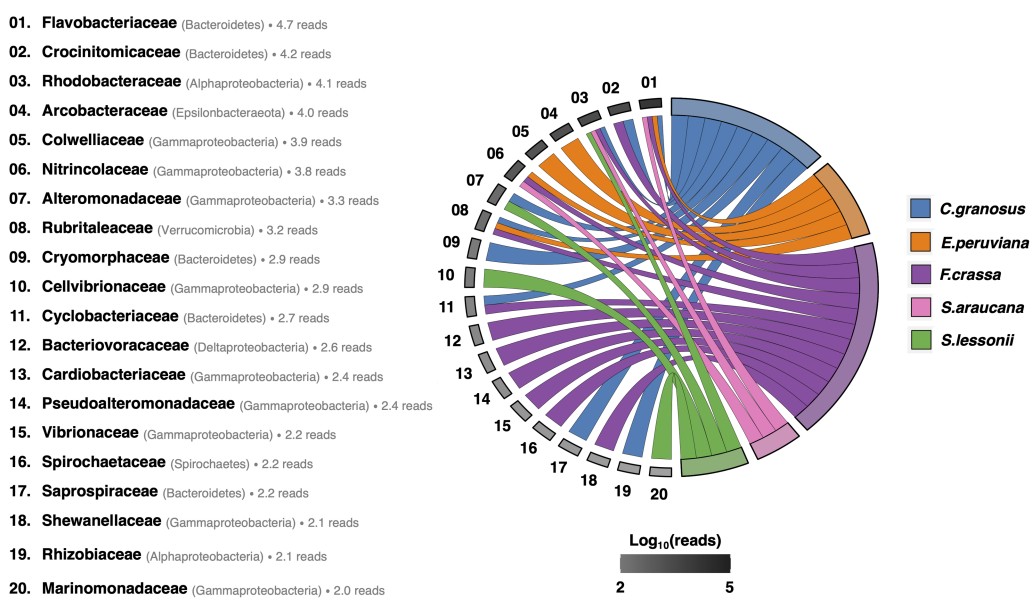

01. **Flavobacteriaceae** (Bacteroidetes) • 4.7 reads
02. **Crocinitomicaceae** (Bacteroidetes) • 4.2 reads
03. **Rhodobacteraceae** (Alphaproteobacteria) • 4.1 reads
04. **Arcobacteraceae** (Epsilonbacteraeota) • 4.0 reads
05. **Colwelliaceae** (Gammaproteobacteria) • 3.9 reads
06. **Nitrincolaceae** (Gammaproteobacteria) • 3.8 reads
07. **Alteromonadaceae** (Gammaproteobacteria) • 3.3 reads
08. **Rubritaleaceae** (Verrucomicrobia) • 3.2 reads
09. **Cryomorphaceae** (Bacteroidetes) • 2.9 reads
10. **Cellvibrionaceae** (Gammaproteobacteria) • 2.9 reads
11. **Cyclobacteriaceae** (Bacteroidetes) • 2.7 reads
12. **Bacteriovoracaceae** (Deltaproteobacteria) • 2.6 reads
13. **Cardiobacteriaceae** (Gammaproteobacteria) • 2.4 reads
14. **Pseudoalteromonadaceae** (Gammaproteobacteria) • 2.4 reads
15. **Vibrionaceae** (Gammaproteobacteria) • 2.2 reads
16. **Spirochaetaceae** (Spirochaetes) • 2.2 reads
17. **Saprospiraceae** (Bacteroidetes) • 2.2 reads
18. **Shewanellaceae** (Gammaproteobacteria) • 2.1 reads
19. **Rhizobiaceae** (Alphaproteobacteria) • 2.1 reads
20. **Marinomonadaceae** (Gammaproteobacteria) • 2.0 reads

*C.granosus*
*E.peruviana*
*F.crassa*
*S.araucana*
*S.lessonii*

$Log_{10}$(reads)

2          5

**Figure 4  Habitat specialist at Family level of the pedal mucus microbiota of grazers.** Habitat specialist at Family level of *Chiton granosus, Echinolittorina peruviana, Fissurella crassa, Scurria araucana,* and *Siphonaria lessonii* pedal mucus microbiota. Phylum or Class names are in parentheses. The grayscale indicates the number of Family reads associated with all the mollusk pedal mucus microbiotas studied; the log number is next to each taxon. The mollusk arcs size shows the number of microbial Families by grazer species.

## Pedal mucus effects on epilithic biofilms

The pedal mucus of both the keyhole limpet F. *crassa* and the pulmonated *S. lessonii* significantly reduced epilithic cover with respect to controls (ANOVA and the Tukey post-hoc test, $df = 5$, $p < 0.001$ and $p = 0.02$, respectively), with the effect of the keyhole limpet significantly higher than that of *S. lessonii* (Fig. 6). The mucus of all other species had no effect over epilithic biofilm (Fig. 6).

## DISCUSSION

### Microbiota and epilithic biofilm microbial communities

We found that the pedal mucus microbiota of all species tested were different from that of the epilithic biofilms, and that they also differed among species. The most similar microbiota were those between the pulmonate limpet *S. lessonii* and that of the periwinkle *E. peruviana*, which both tend to occupy the upper intertidal shore (*Santelices, 1990*; *Hidalgo et al., 2008*). In contrast, the microbiota of the chiton *C. granosus,* keyhole limpet *F. crassa,* and scurrinid limpet *S. araucana* differed markedly from all grazers. Thus, we reject the hypothesis that the same core microbial groups dominating the epilithic biofilms are also dominant in grazer microbiota. Also, we reject that grazers share similar microbial communities in their external microbiota because they all co-occur on the same wave-exposed rocky shore habitat and consume biofilms. Our results are similar with previous studies which found that some invertebrate microbiota were different from

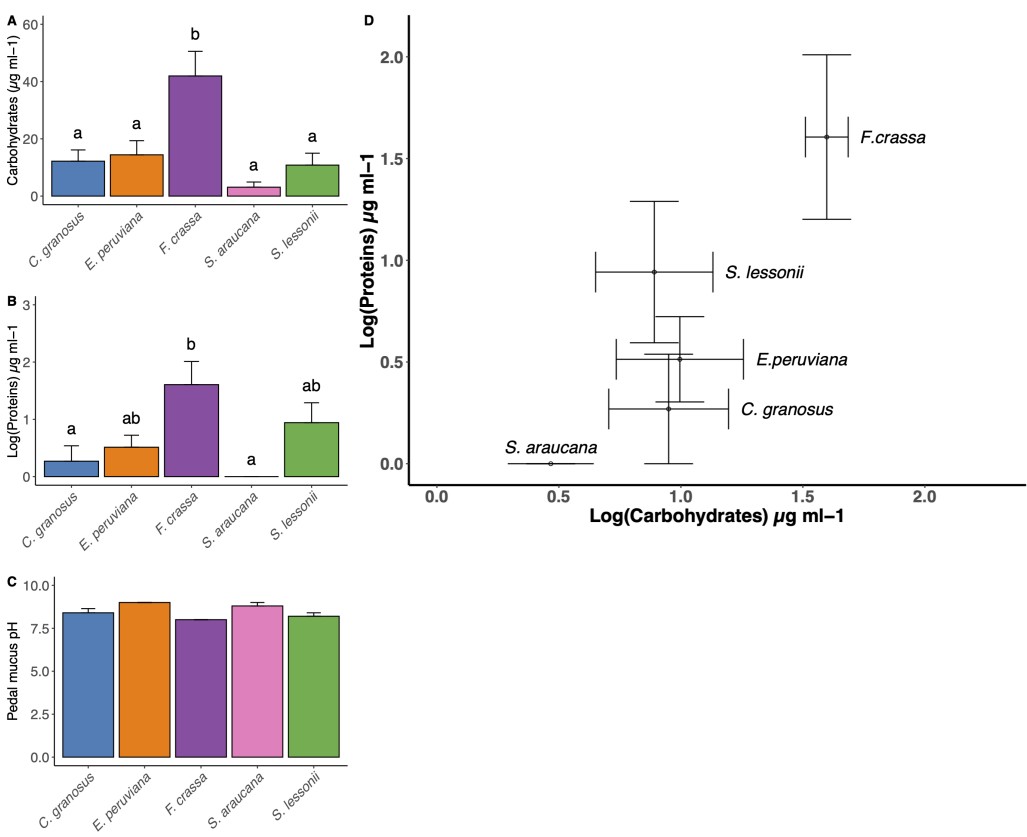

**Figure 5  Pedal mucus Carbohydrates/Proteins content and pH.** Pedal mucus characteristics of *Chiton granosus, Echinolittorina peruviana, Fissurella crassa, Scurria araucana,* and *Siphonaria lessonii.* (A) Carbohydrates content of pedal mucus (B) Protein content of pedal mucus (C) Pedal mucus pH (mean + SE). Different letters above bars indicate significant differences with a posteriori Tukey tests at the experiment-wise error rate = 0.05. (D) Correlation between carbohydrates and protein content of pedal mucus (bars indicated the SE).

the surrounding marine environment (*Lema, Willis & Bourne, 2014*; *Burgsdorf et al., 2014*; *Pantos et al., 2015*; *Kellogg, Goldsmith & Gray, 2017*; *Neu, Allen & Roy, 2019*).

Characterization of host-microbiota is challenging, as it is too easy to include the pool of species present in the natural environment (*e.g.*, seawater) at the time animals are collected as host-microbiota. It is important to identify and remove those species in the surrounding environment when animals are collected from those habitat specialists in the animal's body or the pedal mucus, which requires laboratory acclimations and sterile seawater. However, further research with different approaches is needed to understand and capture the variations of the complete natural microbial communities.

In our study, *E. peruviana* and *S. lessonii,* those with the most similar microbiota co-occur with all other grazer species on wave-exposed rocky platforms (*Broitman et al., 2001*; *Rivadeneira, Fernández & Navarrete, 2002*). Nevertheless, they are found mostly in the upper intertidal zone more frequently than the other grazer species (Fig. 1) (*Otaíza & Santelices, 1985*; *Santelices, Vásquez & Meneses, 1986*; *Aguilera & Navarrete, 2007*). Harsh

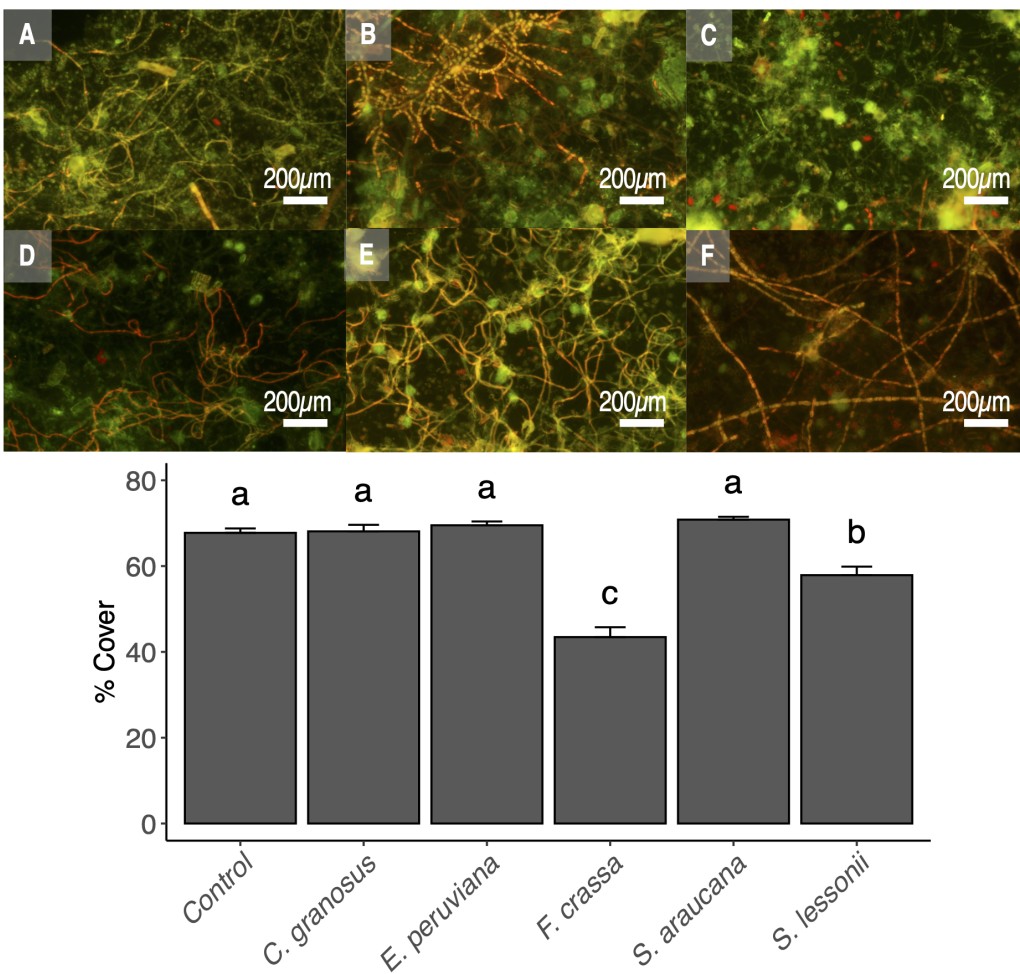

**Figure 6** **Pedal mucus effect over photosynthetic biofilm cover percentage.** Fluorescence microscopy of pedal mucus effect on photosynthetic biofilm cover percentage. Treatments (A) *Control with no pedal mucus,* Pedal mucus of (B) *Chiton granosus,* (C) *Echinolittorina peruviana,* (D) *Fissurella crassa,* (E) *Scurria araucana,* and (F) *Siphonaria lessonii.* Different letters above bars indicate significant differences ($p < 0.05$) among treatments (Tuckey test after one-way ANOVA). Photograph selected from a set of 20 replicates. Photography by Clara Arboleda-Baena.

physical conditions encountered in the upper shore, such as temperature (*Wethey, 1983*; *Williams & Morritt, 1995*; *Finke, Navarrete & Bozinovic, 2007*; *Szathmary, Helmuth & Wethey, 2009*), solar radiation (*Santelices, 1990*; *Huovinen & Gómez, 2011*), and desiccation stresses (*Evans, 1947*; *Stephenson & Stephenson, 1961*; *Lewis, 1964*; *Castilla, 1981*; *Santelices, 1990*; *Harley & Helmuth, 2003*; *Flores, Cienfuegos & Navarrete, 2019*) may influence the composition of the pedal mucus microbiota (higher selection pressure) of these two species, making them more similar. The higher relative abundances of Arcobacteraceae, Alteromonadaceae, Colwelliaceae, Marinomonadaceae, and Saccharospirillaceae (Table S2 in Supplement 3) in the pedal mucus microbiota of *E. peruviana* and *S. lessonii* is not closely related to the environmental factors mentioned above, but further studies should be conducted to evaluate the functional characteristics of these ASVs in the grazers' pedal

mucus. In turn, the less stressful conditions (*i.e.*, lower selection pressure) in the lower shore may permit microbiota to differ more widely among grazer species (*Arboleda-Baena et al., 2021*). A larger variety of species from different shore levels must be investigated to separate effects dependent upon habitat from species-specific effects.

All mollusk species had Flavobacteriaceae followed by the Colwelliaceae (Bacteroidia and Gammaproteobacteria respectively) as the most abundant microorganism sequences (Table S2 in Supplement 3). Bacteroidetes are prone to have a surface-associated lifestyle, supported by the extracellular degradation of complex polymers such as polysaccharides and proteins (*Dang & Lovell, 2016*), like those found in the pedal mucus. Members of Flavobacteriaceae also have been found in the gastrointestinal tract microbiome of other mollusks, such as the blue-rayed limpet *Patella pellucida* (*Dudek et al., 2014*), and are significantly more abundant in the digestive glands of healthy red abalones (*Haliotis rufescens*) than they are in red abalones with infectious diseases (*Villasante et al., 2020*). To the best of our knowledge, ours is the first report of the family Colwelliaceae (Alteromonadales) in Gastropoda and Polyplacophora microbiota. We hypothesize that they could be expected in all mollusks' pedal microbiota because of their capability to hydrolyze organic compounds (*Ivanova, Sébastien & Richard, 2004*) present in the mucus. Alteromonadales have been described in different marine surfaces (*Dang & Lovell, 2016*), like macroalgae (*Egan et al., 2013*; *Neu, Allen & Roy, 2019*) and metal surfaces exposed at sea (*Vasconcelos, 2020*).

The "habitat specialists", ASVs with a higher specificity and fidelity to a specific mollusk microbiota, in *C. granosus, E. peruviana, F. crassa,* and *S. araucana* belonged mostly to Bacteroidetes (Flavobacteriaceae). Specifically, *C. granosus* and *F. crassa* had taxa from Bacteroidetes (Crocinitomicaceae and Cyclobacteriaceae. *C. granosus* also has taxa from Cryomorphaceae and Saprospiraceae). Crocinitomicaceae's (*Muñoz, Rosselló-Móraa & Amann, 2016*) presence could be explained by the amino acids required for their growth (*Bowman, 2020*) which are present in the pedal mucus. This is the first description of this bacterial family in the pedal mucus microbiota of Polyplacophora and *F. crassa*. Moreover, Cyclobacteriaceae (*Nedashkovskaya & Ludwig, 2015*) is widely distributed in diverse marine habitats such as marine surface water, marine organisms, salty water lagoons, oilfield sediments, and solar salterns (*Bhumika et al., 2013*). Members of the family were previously described in the microbiota of the blue-rayed limpet *Patella pellucida* (*Dudek et al., 2014*). Additionally, Cryomorphaceae (*Bowman, Nichols & Gibson, 2003*), also presented in *C. granosus* microbiota, have complex growth requirements necessitating sea-water salts, organic compounds as sole nitrogen sources, yeast extract, and vitamins. Their diversity and particular nutritional preferences suggest a wide range of habitats (*Bowman, Nichols & Gibson, 2003*; *Bowman, 2020*), including the chiton pedal mucus. Saprospiraceae (*Krieg et al., 2012*), for its part, is found in freshwater and/or marine environments, and was previously described in the gut microbiome of the Roswell springsnail (*Pyrgulopsis roswellensis*) and Koster's springsnail (*Juturnia kosteri*) (*Walters et al., 2022*). Their nutritional requirements have not been described yet, and their presence in the pedal mucus requires further research.

Verrucomicrobia ASVs (Rubritaleaceae) (*Hedlund, 2015*) were found to be habitat specialists in *C. granosus, E. peruviana* and *F. crassa* pedal mucus microbiota. The presence of this family in the pedal mucus microbiota may be explained by the oxidation of a wide variety of organic molecules for growth (*Hedlund, 2015*). This Phylum was previously described in the microbiota of the rough periwinkle (*Littorina keenae*) (*Neu, Allen & Roy, 2019*) and in the gut microbiome of the Roswell springsnail (*Pyrgulopsis roswellensis*) and Koster's springsnail (*Juturnia kosteri*) (*Walters et al., 2022*).

Epsilonproteobacteria (Arcobacteraecea), was only found in *E. peruviana* pedal mucus, where it reached high abundance. This class was previously described in other gastropod microbiomes (*Dudek et al., 2014*; *Zbinden et al., 2015*; *Mizutani et al., 2020*). In contrast, *C. granosus, F. crassa, S. araucana* and *S. lessonii* had habitat specialist taxa from Alphaproteobacteria (Rhodobacteraceae). Rhodobacteraceae (*Garrity, Bell & Lilburn, 2005a*), occur in freshwater and marine systems and are frequently associated with biological surfaces (*Dang & Lovell, 2016*). This family has been previously discovered in gills of the giant abalones (*Haliotis gigantea*) (*Mizutani et al., 2020*) and the gut microbiome of the Roswell springsnail (*Pyrgulopsis roswellensis*) and Koster's springsnail (*Juturnia kosteri*) (*Walters et al., 2022*).

Alphaproteobacteria ASVs (Phyllobacteriaceae) were found to be habitat specialists in *C. granosus* pedal mucus microbiota. Phyllobacteriaceae has genera present in marine environments, (*Brenner et al., 2005*; *Liu et al., 2016*) and members of this family were previously described in the gut microbiome of the Roswell springsnail (*Pyrgulopsis roswellensis*) and Koster's springsnail (*Juturnia kosteri*) (*Walters et al., 2022*).

Deltaproteobacteria ASVs (Bacteriovoracaceae) and Spirochaetes ASVs (Spirochaetaceae) were found to be habitat specialists in *F. crassa* pedal mucus microbiota. Bacteriovoracaceae members are aerobic predators of gram-negative bacteria or they can grow saprophytically in a rich nutrient medium (*Davidov & Jurkevitch, 2004*). Their presence in the pedal mucus microbiota exhibits the complexity of interactions found in this environment. Further research is needed to better understand interactions between microbiota members. Members of Deltaproteobacteria were previously found in the blue-rayed limpet (*Patella pellucida*) microbiota (*Dudek et al., 2014*) and the gill cells of the hydrothermal vent gastropod *Cyathermia naticoides* (*Zbinden et al., 2015*). However, this is the first report of Bacteriovoracaceae in mollusk microbiota. The presence of Spirochaetaceae microbiota may be explained by carbohydrates or amino acids required for carbon and energy sources present in the pedal mucus (*Gupta, Mahmood & Adeolu, 2013*), Members of this family have been described in the gills of three abalone species, namely *Haliotis discus, H. gigantea,* and *H. diversicolor* (*Mizutani et al., 2020*).

Pedal mucus microbiota from all grazer species also contained other habitat specialists from Gammaproteobacteria (Alteromonadaceae, Cardiobacteriaceae, Cellvibrionaceae, Colwelliaceae, Marinomonadaecea, Nitrincolaceae, Pseudoalteromonadaceae, She-wanellaceae, and Vibrionaceae) (see Fig. 4). In general, Gammaproteobacteria are surface associated, especially Alteromonadaceae (*Dang & Lovell, 2016*). Nitrincolaceae (Gammaproteobacteria, Oceanospirillales) (*Garrity, Bell & Lilburn, 2005b*) has been discovered on the shell surfaces of the California Mussel (*Mytilus californianus*) in tidepools

and on emergent benches (*Pfister, Meyer & Antonopoulos, 2010*), in the gill microbiome of mangrove lucinids (*Phacoides pectinatus*) (*Lim et al., 2019*), and the large bivalves (*Acesta excavata*) in coral reefs on the northeast Atlantic (*Jensen et al., 2010*). Members of Gammaproteobacteria have been reported as the most plentiful in water and algal samples (*Neu, Allen & Roy, 2019*) and artificial surfaces (*Daille et al., 2020*) in the Southeast Pacific. Regarding mollusk microbiomes, members of this class have been found in the digestive gland of red abalones (*Haliotis rufescens*) (*Villasante et al., 2020*). They are abundant in the gills of the disk abalone *Haliotis discus* (*Mizutani et al., 2020*) and the blue-rayed limpet *Patella pellucida* (*Dudek et al., 2014*). This is the first description of some families in the pedal mucus of Gastropoda and Polyplacophora, however, further research is needed to understand their role in the pedal mucus environment.

## Pedal mucus protein and carbohydrate concentration

According to our results, differences in microbial composition among grazers cannot be explained by differences in the protein/carbohydrate concentrations of their pedal mucus.

The hypothesis that the chemical micro-environment in the pedal mucus determined the composition of the microbial community was therefore rejected. The most similar/different species in terms of microbial community composition were not the most similar/different in mucus composition (compare Figs. 2 and 5). Since grazer species did not exhibit differences in pH, one of the most critical factors in the microbial community structure (*Rousk et al., 2010*), it is possible that our study of the difference in carbohydrates and proteins does not provide sufficient causal explanations for differences in the microbial community. Thus, a study of protein/carbohydrate identities is needed to understand microbial metabolisms and composition (*Martiny, Treseder & Pusch, 2013*). Conversely, our study also suggests that interactions within the microbial community and their host may overcome, to some extent, the conditions imposed by variability in mucus composition. Consequently, further studies should be conducted on factors that could affect the microbiome composition, such as aging, development, diet, and grazer reproduction (*Apprill, 2017*).

## Pedal mucus effects on epilithic biofilms

Our experimental results show that only the mucus of *F. crassa* and *S. lessonii* had significant and negative effects on photosynthetic components of the epilithic biofilm. Since these species tended to have different protein/carbohydrate concentrations than the remainder of the species, especially *F crassa*, we cannot reject the hypotheses that the effects of pedal mucus on the abundance of photosynthetic biofilm are at least partly related to protein/carbohydrate concentration in the pedal mucus. The protein content is correlated with antibacterial effects, both for gram-negative and gram-positive bacteria (*Iguchi, Aikawa & Matsumoto, 1982*), while the carbohydrate concentration is not (*Connor, 1986; Davies, Hawkins & Jones, 1990*). Thus, it appears that the higher protein concentration in *F. crassa* and *S. lessonii* pedal mucus (Fig. 5D) could negatively affect the cover of photosynthetic epilithic biofilms. However, further studies must be conducted to test this hypothesis because our study had a small sample size and other variables could also affect the biofilm cover. Additional studies are needed to investigate the role of protein content in

pedal mucus of different grazers, not only on measures of the photosynthetic components of biofilms, but on the entire biofilm community structure through high-throughput sequencing methods.

## CONCLUSIONS

The process of evolution in co-occurring micro- and macro-organisms allowed many microorganisms to colonize new habitats in the bodies of macroscopic organisms (*McFall-Ngai et al., 2013*), such as the pedal mucus of mollusk grazers. In our results, the pedal mucus microbiota and its carbohydrate/protein content showed variability across grazer species and a potential impact on epilithic biofilms, despite these mollusks co-occurring on the same wave-exposed rocky shore habitat and consuming epilithic biofilms. The differences in microbial composition among grazers and the effects of pedal mucus on the abundance of photosynthetic biofilm were not explained by differences in the overall chemical composition (protein/carbohydrate concentration) of the pedal mucus. Consequently, further studies should be conducted to understand the identities of those proteins and carbohydrates and to analyze other factors that could affect the microbiota composition such as aging, development, and grazer reproduction. Our broad description of the main microbial families found in all (generalists) or some (specialist) of the five-mollusk grazer microbiota is provided here as a first step towards defining a baseline for microbial communities. However, further studies should be done to define ''healthy'' or ''unhealthy'' communities for these grazers. This information would be useful for future assessment of potential responses and alteration due to human-caused perturbations, persistence, and variability of complex microbial assemblages (*Shade & Handelsman, 2012*; *Ribes et al., 2016*), and also to identify core microbes associated with other mollusk hosts under normal or perturbed conditions (*Shade & Handelsman, 2012*), thus providing insight into the ecology and co-evolution of these systems.

## ACKNOWLEDGEMENTS

We are in debt to many students and research assistants at ECIM who collaborated with us in the field and the laboratory, especially Andrés Molina, Bastián Márquez, Florencia Ponce and Paola Vivian.

### Funding

Funding for these studies and for international collaboration was provided by CONICYT–National PhD scholarship Program 2016 and CONICYT (Chilean National Agency for Research and Development (ANID)) international complement to CMAB, and by Fondecyt grants No. 1160289 and 1200636 to Sergio Navarrete, and No. 1171259 to Rodrigo De la Iglesia. Complementary funding was provided by Chilean National Agency for Research and Development (ANID) PIA/BASAL FB0002 to Sergio Navarrete. The small grants for undergraduate research involvement, for winter 2018 and summer 2019, was provided by

the Pontificia Universidad Católica de Chile. The funders had no role in study design, data collection and analysis, decision to publish, or preparation of the manuscript.

## Grant Disclosures

The following grant information was disclosed by the authors:

CONICYT –National PhD scholarship Program 2016.

Chilean National Agency for Research and Development (ANID).

Fondecyt: 1160289, 1200636, 1171259.

Chilean National Agency for Research and Development (ANID): PIA/BASAL FB0002.

Pontificia Universidad Católica de Chile.

## Competing Interests

The authors declare there are no competing interests.

## Author Contributions

- Clara Arboleda-Baena conceived and designed the experiments, performed the experiments, analyzed the data, prepared figures and/or tables, authored or reviewed drafts of the article, and approved the final draft.
- Claudia Belén Pareja performed the experiments, prepared figures and/or tables, and approved the final draft.
- Isadora Pla performed the experiments, prepared figures and/or tables, and approved the final draft.
- Ramiro Logares analyzed the data, authored or reviewed drafts of the article, and approved the final draft.
- Rodrigo De la Iglesia conceived and designed the experiments, analyzed the data, authored or reviewed drafts of the article, and approved the final draft.
- Sergio Andrés Navarrete conceived and designed the experiments, analyzed the data, authored or reviewed drafts of the article, and approved the final draft.

## Field Study Permissions

The following information was supplied relating to field study approvals (i.e., approving body and any reference numbers):

Comité Institucional de Seguridad en Investigación - Pontificia Universidad Católica de Chile approved the study (170829006).

## DNA Deposition

The following information was supplied regarding the deposition of DNA sequences:

The sequence data are available at the European Nucleotide Archive (ENA) database: PRJEB41739.

## Data Availability

The data is available at Figshare: Available at https://figshare.com/projects/Hidden_interactions_in_the_intertidal_rocky_shore_variation_in_pedal_mucus_microbiomes_among_marine_grazers_that_feed_on_epilithic_biofilm_communities_/119655

Arboleda Baena, Clara María (2021): PedalMucusEffect_PhotosyntheticBiofilmCoverPercentage. figshare. Dataset. https://doi.org/10.6084/m9.figshare.15170586.v1

Arboleda Baena, Clara María (2021): PedalMucus_MANOVA_Carbohydrates/ProteinsContent. figshare. Dataset. https://doi.org/10.6084/m9.figshare.15170523.v1

Arboleda Baena, Clara María (2021): PedalMucus_ProteinsContent. figshare. Dataset. https://doi.org/10.6084/m9.figshare.15170520.v1

Arboleda Baena, Clara María (2021): PedalMucus_CarbohydratesContent. figshare. Dataset. https://doi.org/10.6084/m9.figshare.15170514.v1.

Arboleda Baena, Clara María (2021): GrazerAssemblageBiomass.csv. figshare. Dataset. https://doi.org/10.6084/m9.figshare.15113490.v2.

Arboleda Baena, Clara María (2021): PedalMucus_pH. figshare. Dataset. https://doi.org/10.6084/m9.figshare.15115059.v2.

## Supplemental Information

Supplemental information for this article can be found online at http://dx.doi.org/10.7717/peerj.13642#supplemental-information.

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
