# Peer review of "Hidden interactions in the intertidal rocky shore: variation in pedal mucus microbiota among marine grazers that feed on epilithic biofilm communities"

_PeerJ, doi:10.7717/peerj.13642_

## Round 0.1 · original submission · Major Revisions

Thank you for your submission of this very interesting study. We were very lucky to field three excellent reviewers who provided thorough and diverse comments all of which will improve the manuscript significantly. We ask that you carefully address each point in your revision and provide a thorough summary of the response to these suggestions and how you have modified your revised manuscript accordingly.

One minor note - I do acknowledge that there are differences of opinion on whether incidence (binary) vs abundance (proportion) based metrics of communition structure are used in distance-based analyses such as PERMANOVA, nMDS, etc. Reviewer 1 and I have different perspectives on this, as it is my opinion that when using genomic data (in this case amplicons of SSU rRNA genes) it is preferable to, as you have done, use relative abundances (Bray-Curtis dissimilarity) instead of presence-absence (Jaccard dissimilarity) distance metrics, but I respect my colleague's perspective and I hope you can address the idea even if you decide to stick with your Bray-Curtis distance matrices in the analysis.

Reviewer 1 ·

Basic reporting

Title: I would suggest to replace “in the intertidal rocky shore” by “the rocky intertidal”. It´s a bit shorter and keeps the meaning intact.´

The Introduciton is spot on and very well-written. However, I find both the methods and results somewaht hard to read. Below some points that need attention.

Line 171. Not sure “fishers” is correct. Replace by “fishermen”?
Line 172. May be better to spell out 10 to avoid confusion with quadrat size.
Line 175. Are the length-weight relationships derived from published material? If so, provide appropriate citation.
Line 204. I find somehow this sentence read awkwardly “We expected that this cleaning period also helped remove microorganisms not accustomed to growing in the pedal mucus”
Line 205. Here and elsewhere replace “mobility” by “motility”. Mobility is the ability of an object to be moved, whereas motility is the ability of an organism to move independently.
Line 212. “that prevented overheating and potential mineral,” Not sure and potential mineral means…
Line 228. “The highest mean foot area”. The highest or the largest?
Line 230. Remove “besides” as it makes no sense in this context.
Line 381 - Remove comma from after both and lessoni.
Line 384- “The mucus of all other species had no effect with respect to the control”. The sentence reads awkwardly. Do you mean that there was no significant effect of pedal mucus of the other species on biofilm? Re-write.

Experimental design

I find some sections of the methods are hard to follow and do not read fluently.

Was water only sterilized via filtration? Not autoclaved? How effective was this in removing all microorganisms such as bacteria, fungus, etc. Is this why cover glass were used to control for this? If not, it is not clear why they were used.
Line 227. What was the control of epilithic biofilm? This is not described earlier.

Validity of the findings

I am not well versed in molecular analyses so I will rely on other reviewers’ expertise to make a better judgement of the methods used. However, and overall, the methods are quite hard to read. I find the authors make use of a lot of different statistics (PERMANOVA, MANOVA, ANOVA, Welch ANOVA, Linear Discriminant Analysis), different post-hoc tests, etc… Why not keep it simple and use what is standard and well known? The way I see it, most of this could be analysed using standard techniques. There are also some problems with the statistics used:

a) Line 250-251. “microbial beta diversity of the grazer species (i.e., pedal mucus) and epilithic biofilm, we used non-metric multidimensional scaling (NMDS) ordination, based on Bray-Curtis dissimilarities. To test for statistically significant differences in composition among the microbiomes…”. Betadiversity and composition are typically associated with species identities and not abundances, whereas Bray-Curtis is commonly used where abundances data are present. Please justify why Bray-Curtis was used over other incidence-based metrics (e.g. Jaccard).

b) Line 288-289 “, since carbohydrates and protein are correlated, we tested for significant differences among carbohydrate and protein concentration among species using MANOVA,” Correlated data are not supposed to be analysed together. If they were indeed correlated, why not perform ANOVA on only of them? Or at least, test them separately. Also, for simplicity, why not use PERMANOVA as described before instead of introducing yet another (and similar) test such as MANOVA. Keep it simple, and coherent.


Other points:

Line 296. “Welch´s ANOVA” Why not using a regular ANOVA?

Line 350-354 – “We found significant differences in microbiome richness among treatments with similar values among, F. crassa, C. granosus, S. araucana and epilithic biofilms, and significantly lower richness in E. peruviana and S. lesson. The Shannon diversity index also showed significant differences among treatments”. Please provide some figures of the ranges in species richness and diversity. I find it really annoying to be referred to a supplementary file in order to know if species richness ranges were 10, 100 or 1000. Also, this is what results are all about. Not the statistics. Same for the other results (e.g. protein concentrations, etc.)

Additional comments

I have not delved much into the discussion since I do have some reservations regarding the statistics applied (e.g. Bray-Curtis used for composition and betadiversity) and the overall methods and results description which make it hard to fully understand what was done and what the results actually were.

Reviewer 2 ·

Basic reporting

a. The English used throughout needs improvement and the writing, in general, could be more concise and better organized.

Examples of language issues:
First sentence of the abstract, lines 95-96 “also name as”, line 110 “composed by ~96% of water…”, line 112 “Pedal mucus negative effect on”, line 165 “gastropod grazer assemble”, lines 277-278, line 294.

Be careful of tense throughout (e.g., hypotheses in future tense (lines 154-161). These hypothesis can also be more clearly stated.

Sentences (esp. in the introduction) are a bit long with multiple phrases, making it challenging to read. Please revise to make more concise.

The introduction and discussion need to be more concise and better organized—with each paragraph beginning with a strong thesis statement. For instance, starting at line 110, the authors begin talking about the chemical composition of pedal mucus, but the point of the next two paragraphs is that this mucus has been shown to have both negative and positive effects on epilithic biofilms. While the chemical composition is important, it’s not the point of the paragraph—it’s info that should be provided prior to discussing the antimicrobial activity you discuss starting in line 118 OR part of a larger description of pedal mucus, which is lacking (and I mention more about below). In the discussion, an example is that the first sentence needs to be broken into two thesis statements, each followed with the supporting results.

b. Additional background and context are needed:

Lines 102-104. While it is technically correct that the mucus is in contact with the rock, it’s really the foot of the mollusc, no? Consider the wording here. Define pedal mucus for the reader who may not work on mollusks, inverts, etc. Just a few words re: where it’s secreted from, its function to make this work more accessible to a broader audience.

Lines 105-109. This sentence should be incorporated into another paragraph, perhaps the previous?

Line 113: Achantina should be Achatina

Line 134. I’m not sure why carnivorous molluscs are mentioned here? More context is necessary.

Lines 149-150. How is this different from what you are saying in lines 125-126? B/c in Chile? Please clarify.

c. The manuscript uses standard sections, but the abstract is missing the Background section, a meaningful objective, and the wording needs to be more intentional and concise.

Taxonomic authors of Linnean binomials are not provided for the molluscs.

d. The figures are relevant and of high, visual quality, but there are some issues that need to be addressed. The raw data have been appropriately shared.

Figure 1: I like this figure, but it’s difficult to tell which mollusk image belongs to which bar. Also, are these images taken at the same scale? A scale bar needs to be included. Why not just show images of the species used in this study?

Should the headings read “intertidal” instead of “tide”.

Change y-axis label to g m-2

Figure 3b: The yellow is a little hard to see

Figure 4: This figure looks nice, but when you break it down, it’s a bit hard to interpret. I think most could be clarified with a more detailed legend and some adjustments:

The gray scale is hard to see—It appears to increase as you go clockwise, but I can only differentiate 3 colors. I wonder if a legend (like the one you have for the mollusk species) would be more suitable if there are only 3 or 4 different grayscale colors?

Also, if there are multiple mollusk species whose mucus was composed of seqs from the same family (e.g. Flavobacteriaceae), the log (read) for each of those was the same or that’s the total with 1/3 of the reads = Flavobacteriaceae from C. granosus, 1/3 from F. crassa, 1/3 from S. araucana? Not sure how to make this clear to the reader unless you add text to the legend.

Do the size of the arcs mean anything for the mollusk species (other than composition of 3 vs 4 families)?

Italicize and spell out species names in legend.

Fig. 4 legend: Squares unnecessary. I would add something like, “phylum names in parentheses” (and put them in parentheses in the figure itself). Legend is incomplete: finish the last sentence “reads associated with….”

Fig. 5 legend: Needs to be edited carefully. Also need to include in the legend what the letters above the bars mean.

Fig. 5: axis labels for a, b, d need to be ug ml-1. Would be nice to continue color scheme from previous figures.

Fig. 5c: “mucus” should be added to the y-axis label to avoid confusion

Fig. 6 legend: Again some typos. Would also make sure to include the replicate number and that these images are from one of the reps. Need to state these are means plus/minus SE? Photographer identities missing.

Experimental design

a. This is original primary research is within the aims and scope of PeerJ.

b. The authors do a good job pointing out why the study is important: most research has focused on trophic interactions and internal microbiomes compared to non-trophic/external interactions. Further, we don’t know much at all about pedal mucus microbiomes.

I do not think the research question is well-defined, however. In line 151-153, the authors say that they “developed a method to characterize the microbiomes…” but I do not believe this is what they mean because the methods describe standard amplicon high-throughput sequencing/metabarcoding, which has already been developed. I think a clearly stated objective right before stating the hypotheses would be very helpful to the reader and a better description of how they plan to address each question.

For the first hypothesis, you give a nice reason for why you think all the communities will be homogenous. Please lay out your reasoning for hypotheses 2 and 3. For the second hypothesis: I’m not sure I understand why the chemical composition of the pedal mucus would only affect “small” differences seen?

c. I am concerned about the number of reads that were generated and used in subsequent data analyses. In line 248, the authors say they rarefied the data to 1869 reads/sample. Did the rarefaction curves plateau at such a low read #/sample? Did you run these samples with other projects? A 2 x 300 bp run should yield ~20 million reads, so with 30 samples, you’d expect something like 600,000 reads/sample. Please clarify.

d. The methods used are appropriate and of a high standard, and the authors described most of the methods well, but I’ve asked for some clarification below.

Line 175: Did you sum across all three zones? Please clarify. It’s hard to tell which species are the top 5 most abundant in Figure 1 b/c of the way the figure is broken into the 3 zones surveyed.

Line 186: Which zone were the mollusks collected for this portion? All 3?

Line 189-190: sentence fragment

Line 193: why mentioning supplementary 2 here? Also supplementary 1 isn’t mentioned until line 228, so there are some issues w/ order and relevancy that need to be addressed.

Lines 198-199, 208-209: what salinity? Same as that of their sampling location?

Line 205: maybe state here “Six treatments with 5 replicates each”

Line 214: Please explain more about glass slides—size? were they submerged in water? Were the snails placed directly on these slides? Could snails go elsewhere in the aquaria? Include words “for pedal mucus collection” as well.

Line 216: should be d, e ,f (no g)

Lines 216-221: I’m confused by what is meant by “a single common reference”? Can you please clarify?

Line 221: random assignment in the facility’s space?

Line 226-227: how were the replicates lost? Please explain.

Line 226: not sure the n=5 is appropriate here

Line 227: typo

Were the snails’ shells scrubbed at any point to avoid contamination from this potential source?

Line 262: typo
Line 266: typo

Line 286-287: n=5s necessary here?

Line 295: another unnecessary placement of “n=5”?

Line 324: This sentence needs to be rewritten for clarity.

Line 330: Need to make clear that you are discussing sequences and not the actual specimens. So: “Sequences from more than 17 phyla.” Similarly throughout this entire section: e.g., Verrucomicrobia sequences were the most relatively abundant (lines 331-332).

Line 354: I don’t see Supplementary 7 Figure S1b. Do you mean Supplementary 6 Fig S1b? Check figure and table citations carefully.

Line 359: “Flavobacteriaceae sequences.”

Lines 359-367: Edit to refer to the sequence composition. How can an ASV can be a “strict specialist.”

Line 370-371. According to Fig. 5a, though, it appears that only F. crassa differed w/ re: to carbs, so this statement does not agree with the figures.

Line 374: acronym already described in methods. Perhaps remind readers of why you did the LDA.

Line 377: Pedal “mucus” pH, yes? I think you could also remove the “around” and be precise here.

Line 378: I think Fig 5c could be cited after “among species” at the end of the sentence.

Validity of the findings

a. All underlying data have been provided. They appear to be statistically sound, but my concern is that diversity may be underestimated due to low number of sequence reads/sample used in the microbiome data analyses (see comment above).

b. As stated above the discussion needs improvement with re: to organization. It is also missing links back to the introduction and results.

Line 391: both of which or just the periwinkle?

Lines 393-396: this sentence is too long.

Line 396: “These results” – use “Our” to avoid confusion.

Line 397-399: which inverts and in which environments? Still marine?

Line 399-402: This sentence seems to contradict the previous sentence.

Lines 411-413: redundant w/ line 391

Line 414: Should “stresses” be moved to after “solar radiation” in line 417?

Line 419: higher RELATIVE abundances

Lines 419-424: Do you mean to say that these families are not known to be associated with these factors? Or we already know they are not closely related to these factors?

This is the first mention of these families being in higher relative abundance for these species—I don’t believe it’s mentioned in the results. Why not? Did you also find that these were statistically higher (or just appear higher when looking at the graphs)?

Did you dig down into these ASVs to see which genera or species might be present? Maybe more answers here? For example, did you find any of the species mentioned in lines 115-118 of the intro?

Line 421: first mention of Supplementary 4 – need to reorder.

Line 425: not sure that “benign” is the right word. Perhaps the environmental conditions are less variable? Do you have a citation for this?

Have other microbiome studies found differences among intertidal zones whether in the environment or within a species that spans the entire intertidal? I like that figure 1 is broken into the 3 different zones (providing more information to the reader), but differences b/t zones is not mentioned.

Line 428: Again, be careful not to equate sequences with the organisms themselves.

Line 429: This supplemental table is not relevant here.

Line 454, 458: Placement of references not appropriate.

Line 459 and 461: shell surfaces?

Lines 462-464: do we know the role of Nitrincolaceae in these shell surface & gill environments of these other organisms? Or do these citations suggest what their role might be? (same question for next paragraph)

Can you discuss more about what it means that they appear to have these strict specialists?

Lines 466-469: Sentence needs to be rewritten.

Lines 490-492: and also diet, yes?

Additional comments

Throughout the MS, Siphonaria lessonii is missing the last “i” in the species name.

Placement of internal citations need to be revisited throughout (several examples listed above).

Reviewer 3 ·

Basic reporting

This work examined mucus microbiome compositions of five marine mollusk grazers, mucus carbohydrate and protein concentrations, as well as how the mucus impact biofilm microbial communities. It provides an initial assessment of mucus microbial diversity associated with the five mollusks, as well as potential ecological implications. The manuscript is relatively well-written with a comprehensive introduction and a good discussion.

The language could be more concise and precise throughout. For example, the introduction uses phrases like "wonderful exceptions" or "wonderfully complex". Those don't add to the content and in my opinion are not necessary.

Experimental design

The experiments appeared to be well-designed and methods were described in details. The DNA extraction, sequencing, statistical analyses, mucus protein and carbohydrate concentration assessment, and mucus effects on biofilms were well written. However, the description of pedal mucus microbiome collection section is confusing at times.

Line 29: It stated that "six treatments were applied". I don't understand what the "treatments" are. Are those natural biofilms plus the five mollusk species? Why is cover glass slide listed as a treatment? What are the purpose of the letters a), b), c), etc. ? The letter e) is missing ad they don't add up to six. Overall, this section needs a much clearer description.

Line 221: "Treatments were randomly assigned to 30 experimental unites". What does this mean? There are 5 replicates per "treatment"?

Line 223: "rocks and slides were sonicated". Maybe this is standard practice, but I wonder how effective this method is for collecting pedal mucus. How do we know the end product is mainly composed of mucus microbiomes?

Validity of the findings

Several points in the results need more clarification

Line 342: "with relatively homogenous composition within F. crassa and C. granosus, and a large dispersion in S. araucana ".  What does homogenous mean here? You mean within species variations are low?

Line 359: What is the definition of strict specialists? Is it just ASVs only found in one mollusk species? I found the term "specialists" a little confusing, because it implies ecological functions.

Figure 3: Why do we need two separate figures for the NMDS? Isn't 3B capturing all the groups?

Figure 4: the legend uses the same black squares for all species. Since the species are distinguished by color in the actual figure, there is not need to list the square shape in the legend.

A few points in the discussion requires more elaboration:

Line 405: Again, what is the implication of some ASVs being "specialists"? I'm hoping to see some more discussion on the origin and functions of these specific lineages. If some of the lineages are not found in the surrounding environment, where do the mollusk host obtain these taxa? In particular, it was mentioned later that the "specialists" taxa may also occur in other freshwater and marine systems (line 454). Then why are they considered "specialists"?

Line 452: In seems to be implied here that the microbiome is relying on the mollusk mucus for nutrition? Can you elaborate how the mollusk hosts ecology may impact this aspect (i.e. different mucus nutritional value)?

---

## Round 0.2 · Major Revisions

This manuscript continues to be challenging. I concur with many of the issues raised by reviewer 1, and we have been unable to get additional reviewers to address the split decision of the reviewers.

I request that the authors consider the extensive ongoing recommendations of Reviewer 1, which I agree with thoroughly, and revise the manuscript better to ensure these are addressed in a final revision for consideration. In the meantime we will seek additional reviews.

One key issue raised by R1 is rarefaction. Your supplemental table indicates that it is really the epilithic biofilms which have so few sequences and mandate a common denominator of <2000 reads. I encourage the authors to address this issue by re-analyzing the dataset using a rarefaction level of 10,000 reads but excluding the Epilithic Biofilm samples altogether. This allows a deeper analysis, and should be presented as supplemental and analyzed to demonstrate that patterns among pedal mucus microbiomes are robust to deeper sequencing (and not artifacts of the shallow sequencing/rarefaction level mandated by the epilithic biofilm samples).

Reviewer 2 ·

Basic reporting

Original comment: The English used throughout needs improvement and the writing, in general, could be more concise and better organized.

A/ We made an effort to make presentation more straightforward. Thanks very much for the specific suggestions.

Reviewer Response: The authors made attempts to improve the English throughout, paying specific attention to those issues that the reviewers pointed out, but there are still areas where the language needs improvement. There are issues especially where additional text was added/replaced (e.g., the hypotheses at the end of the introduction need special attention and there are still issues with tense throughout).

Issues where text is confusing:

Abstract: “The microbiomes of the chiton Chiton granosus, keyhole limpet Fissurella crassa, and scurrinid limpet Scurria araucana differed markedly.” From one another or from the pulmunate limpet and periwinkle?

Line 88: Not clear if the authors are talking about the interactions or the inverts and macroalgae themselves being studied for over a century, esp. b/c in next sentence oldest ref is 1977.

Lines 93-96: These sentences still need clarification. “Some of” what have no similar counterpart? Interactions? But the microbes are interacting w/ macroscopic organisms in the next sentence so I’m not sure what point the authors are making here.

Lines 104-110: The information added is very helpful, but needs some wordsmithing. Here’s my attempt:

“Pedal mucus, which is essential for animal motility in all mobile gastropods and chitons (REF) and is secreted by the pedal gland located inside the front end of the foot (Davies & Hawkins, 1998), is in direct contact with the rock surface and biofilm microbial communities, yet its microbiome has not been studied in detail.”

Line 118: Pedal mucus “can have” instead of “has”

Original comment: The introduction and discussion need to be more concise and better organized—with each paragraph beginning with a strong thesis statement.

A/ We reorganized the introduction and we thoroughly revised the Discussion and added subtitles to improve the connection with results.

Reviewer response: The introduction is better organized and flows nicely.

Original comment: Taxonomic authors of Linnean binomials are not provided for the molluscs.

A/ Yes. Now we provided. Thank you:
“We chose five of the most abundant species in terms of total biomass (Fig. 1): one Polyplacophoran, the chiton Chiton granosus (Frembly, 1828) Family Chitonidae, and four Gastropods, the Littorinid Echinolittorina peruviana (Lamarck, 1822) Family Littorinidae, the keyhole limpet Fissurella crassa (Lamarck, 1822) Family Fissurellidae, the scurrinid limpet Scurria araucana (d’Orbigny, 1839) Family Lottiidae, and the pulmonate limpet Siphonaria lessoni (Blainville, 1827) Family Siphonariidae..”

Reviewer Response: Authorities are not cited like references. The parentheses have a specific meaning for authorities (i.e., that the species was originally described under a different name, so synonyms exist). If no parentheses = the species has had only one scientific name (the case for C. granosus). If including authorities, please make sure this is done properly.

I suggest using this format for including the family names: “Chiton granosus Frembly, 1828 (Chitonidae), and four gastropods...”

Original comment: The figures are relevant and of high, visual quality, but there are some issues that need to be addressed.

Figure 1: I like this figure, but it’s difficult to tell which mollusk image belongs to which bar. Also, are these images taken at the same scale? A scale bar needs to be included. Why not just show images of the species used in this study?

A/ OK. The scale bar can´t be seen well. Then, we modified the size of the animal photographs to the proportion observed in the intertidal rocky shore. We only show images of the species used in our study. Thank you.

Reviewer response: Figure 1 is not as busy now that it only includes pictures of the species the authors focused on in this study, but the pictures still detract from the data. I think there should be a separate figure (or a part B of figure 1) showing pictures of these species with appropriate scale bars. Scale bars aren’t helpful if we can’t see them!

I’m also unsure why the authors include species that were in very low/0? biomass (e.g., Scurria spp.)—removing these would improve the figure. Perhaps just put a table in the supplement detailing the other species found and only include the species of interest in this bar chart? Or if you decide to keep all species found, bold the species of interest on the x-axis.

Original comment: Figure 3b: The yellow is a little hard to see
A/ OK. We changed the color, thank you.

Reviewer response: Thank you for improving the color-scheme.

Given that the Jaccard and B-C plots are identical (except axis scales), I think the authors only need to present one of these plots. If the other reviewers think it’s necessary to include the results of the Jaccard Distance analysis, then I would say the authors could simply mention in the text that the Jaccard results were qualitatively similar to the B-C results.

Experimental design

Original comment: I do not think the research question is well-defined, however. In line 151-153, the authors say that they “developed a method to characterize the microbiomes…” but I do not believe this is what they mean because the methods describe standard amplicon high-throughput sequencing/metabarcoding, which has already been developed.

A/ Ok. We changed and revised the text:
“In this model ecosystem, we characterized the microbiomes of five common but distantly related intertidal species of central Chile, all of which feed on biofilms, as a first step to understand potential non-trophic interactions between grazers and epilithic biofilms.”

Reviewer Response: Thanks, this is better. I’m not sure it’s important to point out that the mollusks are biofilm feeders here, given your focus on non-trophic interactions. Okay to include in their overall description, of course. How about:

In this model ecosystem, we characterized the pedal mucus microbiota of five common, but distantly related, intertidal mollusks of central Chile as a first step in understanding potential non-trophic interactions between grazers and epilithic biofilms.

Original comment: For the first hypothesis, you give a nice reason for why you think all the communities will be homogenous. Please lay out your reasoning for hypotheses 2 and 3.
A/ Ok, done:
“2) since the intertidal grazers species may have differences in the pedal mucus content of protein/carbohydrates , differences in microbial composition among grazers, and 3) in the effects of pedal mucus on the abundance of photosynthetic biofilm will be primarily explained by differences in the overall chemical composition of their pedal mucus.”

Reviewer Response: As stated above, this still needs some wordsmithing.

Original comment: I am concerned about the number of reads that were generated and used in subsequent data analyses. In line 248, the authors say they rarefied the data to 1869 reads/sample.
Did the rarefaction curves plateau at such a low read #/sample? Did you run these samples with other projects? A 2 x 300 bp run should yield ~20 million reads, so with 30 samples, you’d expect something like 600,000 reads/sample. Please clarify.

A/ Yes, the number of reads used for the analyses were not the optimal but are acceptable. We rarefied the data to 1869 reads/sample due to the size of the smallest dataset from one replicate of the epilithic biofilm control. Additionally, we presented in Supplementary 1 the table with the number of ASVs before and after rarefaction to verify this. Thank you. We clarified this in the manuscript:

“Rarefaction curves were generated with a fixed sampling effort of 1,869 reads per sample due to the size of the smallest dataset from one replicate of the epilithic biofilm control (Supplementary 1).”

Reviewer Response: What do the authors mean by “acceptable”? By what standard? My concern is that the rarefaction curves for some of the samples may still be increasing at depth = 1869 reads/sample and that all samples may not be increasing at the same rate. For instance, average # of ASVs for S. araucana pedal mucus is reduced almost by half when rarefied (42%; Supplemental Table 1), and this % decrease varies greatly across all samples, ranging from 10% to 57%. I would like to see the rarefaction curve, because I can imagine that the resulting data may change at higher rarefaction depths.
In the first draft, the authors did show an NMDS plot that did not include the biofilm control. Were these data rarefied to the fewest # of sequences from that dataset (n = 10599)? If so, then maybe there is no problem as the NMDS looked the same as when the authors included the control, but I suspect these data were rarefied at n = 1869. I suggest examining the data at this higher depth (esp. if the rarefaction curves are closer to or are plateauing here).

Another approach would be to analyze the data w/out rarefaction (so as not to lose so many sequences. Or try to do a variance stabilizing transformation to normalize their data (McMurdie & Holmes 2014; https://doi.org/10.1371/journal.pcbi.1003531).

I also still have concerns that something also went wrong with the sequencing run itself given the low number of reads/sample overall and that the authors lost some of the replicates (“During the microbiome analysis, we lost one sample of each of the following treatments due to a poor-quality Illumina sequencing run.”). I’ll ask again, did the authors include other samples on this run and thus had fewer reads/sample than expected?

New comment: I overlooked this before, but did the authors run PERMDISP (betadisper in R) to ensure that assumptions for the PERMANOVAs were being met?

Original comment Line 186: Which zone were the mollusks collected for this portion? All 3?

A/ Yes. The tidal zone for collection varied among species, focusing in the zone where they were most abundant. You can see this in Figure 1.

Reviewer response: Please be sure to mention this in the methods. I didn’t see any additional text explaining this point.

Original comment Lines 198-199, 208-209: what salinity? Same as that of their sampling location?
A/ Yes, we took the animals and the seawater from the same location, the Estación Costera de Investigaciones Marinas of the Pontificia Universidad Católica de Chile. We clarified this in the manuscript. Thank you:

“…and 0.2 µm filtered seawater (took it from the same location).”

“…five grazers of each species were chosen randomly and placed in individual clean aquaria (14.3 x 14.3 x 12.5 cm) with 400 ml sterile seawater (0.22 µm filtered and took it from the same location).”

Reviewer Response: I saw reviewer 1’s comments re: sterilization. I wouldn’t consider 0.22 um-filtered seawater to be sterile, so please correct language throughout the methods to say that it was “filtered seawater” unless it was actually sterilized via autoclave or UV.

Original comment Line 205: maybe state here “Six treatments with 5 replicates each”

A/ OK:
“Six treatments (a-f) with 5 replicates each were applied:…”

Reviewer Response: Seeing the other reviewers’ comments, I agree that this should be five treatments with 5 replicates each and one control also with five reps. I would mention the control last as you describe the experimental set up.

Original comment Line 214: Please explain more about glass slides—size? were they submerged in water? Were the snails placed directly on these slides? Could snails go elsewhere in the aquaria? Include words “for pedal mucus collection” as well.

A/ Yes, we rewrite this.
“For mollusk pedal mucus collection, the other five treatments had cover glass slides of 75 x 25 x 1.5 mm previously autoclaved and placed in the bottom of aquaria.”

The animals could not go elsewhere in the aquaria, but we think this does not need to be explained.

Reviewer response: Apologies for not understanding, but the aquaria are so much larger than the slides, so I am having trouble visualizing how you could restrict the mollusks' movement to the small slide.

Original comment: Were the snails’ shells scrubbed at any point to avoid contamination from this potential source?

A/ Yes, the entire animal was washed with the autoclaved seawater:
“To minimize contamination from grazers’ feces, two days before the experiment, individuals were “cleaned” by placing them in an aquarium with constant aeration and 0.2 µm filtered seawater (took it from the same location). The two days period was long enough to reduced feces during the experiment and short enough to avoid locomotory and metabolic adverse effects (Calow, 1974), because on the third day animals reduce their motility (personal observations). The sterile water was replaced every 2-6 hours to minimize ammonium concentration and prevent biofilms formation associated with animals´ feces. We expect that this cleaning period also helped remove microorganisms that are incidental on the animal foot and do not maintain populations in the pedal mucus.”

Reviewer response: But there was no physical scrubbing of the mollusks?
Do you mean autoclaved in your response here or filtered? The text states “filtered.”

Original comment: Line 324: This sentence needs to be rewritten for clarity.

A/ Ok. Done. Thank you:
“We obtained 707,318 good-quality sequences from 26 samples. After rarefying to 1,869 reads per sample, due to the size of the smallest dataset from one epilithic biofilm sample, we got a total of 958 ASVs from all treatments (Supplementary 1),….”

Reviewer response: Supplemental table 1 totals to 683,494 sequences so there is an issue here.
Also the # of ASVs from all treatments cannot be found in Supplemental Table 1, only the # of ASVs/sample.

Original Lines 359-367: Edit to refer to the sequence composition.
A/ Ok. Done. Thank you:
“C. granosus also has habitat specialists from Bacteroidetes (Crocinitomicaceae and Cryomorphaceae) sequences. In turn, F. crassa and S. araucana have habitat specialists from Alphaproteobacteria (Rhodobacteraceae) and Gammaproteobacteria (Nitrincolaceae) sequences. In the case of E. peruviana and S. lessonii, the most abundant habitat specialist ASVs were Gammaproteobacteria (Colwelliaceae). E. peruviana also had habitat specialist ASVs from Gammaproteobacteria (Nitrincolaceae) and Epsilonproteobacteria (Arcobacteraceae); whilst S. lessonii had ASVs from Gammaproteobacteria (Alteromonadaceae, Cellvibrionaceae and Marinomonadaecea) sequences (Fig. 4, Supplementary 7).”

Reviewer response: The wording here still isn’t quite right. An example that might help with rewording:

Alphaproteobacteria ASVs were found to be habitat specialists in F. crassa and S. araucana pedal mucus.

Validity of the findings

Original comment: All underlying data have been provided. They appear to be statistically sound, but my concern is that diversity may be underestimated due to low number of sequence reads/sample used in the microbiome data analyses (see comment above).

A/ It is true we maybe underestimating total richness. However, if all the samples were rarefied to the same number of sequences and were compared with each other, the same proportion of differences observed would be detected in the hypothetical case of having more sequences. This is the basis of rarefaction, which attempts to standardize sampling effort and compare samples in a better way. Additionally, we presented in Supplementary 1 the table with the number of ASVs before and after rarefaction to verify this.

Reviewer response: Please see comments above. Because of this matter, I have only scanned the discussion revisions as I'm not sure if the results will remain the same given reanalysis of the data.

Additional comments

I would argue for replacing microbiome with microbiota in the title (and throughout the MS where appropriate) as this is a metabarcoding study and not a metagenomic study (apologies for not mentioning this in the first round of comments).

Watch the spelling of Tukey

S. lessonii in some places still misspelled

Reviewer 3 ·

Basic reporting

The authors have clarified the confusion in the method section and addressed my other concerns in the discussion.

Experimental design

The mucus collection method is now clarified.

Validity of the findings

The revision of methods and figures are good. It seems that the new addition using Jaccard distance didn't change the result. I wonder if only one NMDS figure is enough to present. You can just add that the Jaccard distance analyses revealed the same pattern. Optional, but may make the presentation cleaner.

---

## Round 0.3 · Minor Revisions

We have reviewed your manuscript and with one additional round of edits and improvements, we believe it will be acceptable for publication in PeerJ. Please submit a response to reviewers addressing the final comments provided. Also, please have a professional editing service address the final issues with English grammar.

Reviewer 2 ·

Basic reporting

There are still issues with grammar and typos (especially in the Discussion and Conclusions), but these can be easily addressed by an editor. The manuscript is much improved.

One major issue: Hypothesis #3 in the Intro needs to be a full-thought.

Minor issues:

Line 452: differed markedly from what? Not ALL of these from one other and not ALL with S. lessonii & E. peruviana b/c of S. araucana overlapping with the latter & w/ C. granosus. Needs clarification.

Line 463: do you mean autoclaved or sterile instead of filtered here?

My responses to specific comments from previous reviews:

Lines 93-96: These sentences still need clarification. “Some of” what have no similar counterpart? Interactions? But the microbes are interacting w/ macroscopic organisms in the next sentence so I’m not sure what point the authors are making here.
R/ Yes, "some of" is related to interactions. We have clarified this in the text.
The authors try to point out that those kinds of interactions between micro-and macroorganisms do not have a similar example between macroscopic organisms. Have we clarified your doubt?
ROUND 3: Yes, this is much clearer.


4.1 Figure 1: I like this figure, but it’s difficult to tell which mollusk image belongs to which bar. Also, are these images taken at the same scale? A scale bar needs to be included. Why not just show images of the species used in this study?
A/ OK. The scale bar can´t be seen well. Then, we modified the size of the animal photographs to the proportion observed in the intertidal rocky shore. We only show images of the species used in our study. Thank you.
Reviewer response: Figure 1 is not as busy now that it only includes pictures of the species the authors focused on in this study, but the pictures still detract from the data. I think there should be a separate figure (or a part B of figure 1) showing pictures of these species with appropriate scale bars. Scale bars aren’t helpful if we can’t see them!
I’m also unsure why the authors include species that were in very low/0? biomass (e.g., Scurria spp.)—removing these would improve the figure. Perhaps just put a table in the supplement detailing the other species found and only include the species of interest in this bar chart? Or if you decide to keep all species found, bold the species of interest on the x-axis.
R/ We changed the photos ubication, and we included the scale bar.
Regarding removing grazers species from the graph, we did not because, in this figure, we showed all species with biomass > 0 across the three intertidal zones (See Table GrazerAssemblageBiomass_PeerJ.csv in Figshare https://doi.org/10.6084/m9.figshare.15113490.v2). But we followed the reviewer's suggestion to bold the species of interest on the x-axis and put it in color. Thank you.
ROUND 3: This looks great. Legend: might want to add “Species selected for analysis are highlighted…”

Experimental design

The authors have sufficiently addressed my concerns re: the low read numbers. Thanks to the authors for redoing the analyses at a higher rarefaction depth (where it appears that all samples have reached a plateau) and adding the PERMDISP results.

Validity of the findings

No Comment.

---

## Round 0.4 · accepted · Accept

Thank you for your steadfast dedication to improving this manuscript. We are excited to be able to publish it in PeerJ!